



# Assessment of the Finite VolumE Sea Ice Ocean Model (FESOM2.0), Part II: Partial bottom cells, embedded sea ice and vertical mixing library CVMIX

Patrick Scholz[1], Dmitry Sidorenko[1], Sergey Danilov[1,2], Qiang Wang[1], Nikolay Koldunov[1], Dmitry Sein[1,4], Thomas Jung[1,3]

[1] Alfred Wegener Institute Helmholtz Center for Polar and Marine Research (AWI), Bremerhaven, Germany
[2] Jacobs University Bremen, Department of Mathematics & Logistics, Bremen, Germany
[3] University of Bremen, Department of Physics and Electrical Engineering, Bremen, Germany
[4] Shirshov Institute of Oceanology, Russian Academy of Science, 36 Nahimovskiy Prospect, Moscow, Russia 117997

Correspondence to: Patrick Scholz (Patrick.Scholz@awi.de)

## Abstract

The second part of the assessment and evaluation of the unstructured-mesh Finite-volumE Sea ice-Ocean Model version 2.0 (FESOM2.0) is presented. It focuses on the performance of partial cells, embedded sea ice and on the effect of mixing parameterisations available through the CVMIX package.

It is shown that partial cells and embedded sea ice lead to significant improvements in the representation of the Gulf Stream and North Atlantic Current as well as the circulation of the Arctic Ocean. In addition to the already existing Pacanowski and Phillander (fesom_PP) and K-profile (fesom_KPP) parameterisations for vertical mixing in FESOM2.0, we document the impact of several mixing parameterisations from the Community Vertical Mixing (CVMIX) project library. Among them are the CVMIX versions of Pacanowski and Phillander (cvmix_PP) and K-profile (cvmix_KPP) parameterisations, the tidal mixing parameterisation (cvmix_TIDAL), a vertical mixing parameterisation based on turbulent kinetic energy (cvmix_TKE) as well as a combination of cvmix_TKE and the recent scheme for the computation of the Internal Wave Dissipation, Energy and Mixing (IDEMIX). The IDEMIX parameterises the redistribution of internal wave energy through wave propagation, nonlinear interactions and the associated imprint on the vertical background diffusivity. Further, the benefit from using a parameterisation of sea ice melt season mixing in the surface layer (MOMIX) for reducing Southern Ocean hydrographic biases in FESOM2.0 is presented. We document the implementation of different model components and illustrate their behaviour. This paper serves primarily as a reference for FESOM users but is also useful to the broader modelling community.

## 1    Introduction

Global unstructured-mesh ocean models start to be widely used in climate studies, including the recent CMIP6 simulations (Semmler et al., 2020), although structured-mesh ocean general circulation models are still more mature in terms of features, functionality and complexity due to their long development history. However, step by step, also the unstructured-mesh ocean models acquire new features and catch up in their



functionality. This paper continues the work by Scholz et al. (2019) in documenting the features available in
Finite volumE Sea ice Ocean Model version 2.0 (FESOM2.0, Danilov et al., 2017). It focuses on two
aspects. The first one is about partial bottom cells and embedded sea ice, both of which essentially rely on
the Arbitrary Lagrangian Eulerian (ALE) vertical coordinates used in FESOM2.0. The second one deals with
mixing parameterizations enabled through the use of Community Ocean Vertical Mixing (CVMIX, Griffis et
al. 2015, Van Roekel et al. 2018) package.
Partial bottom cells were first introduced for a finite volume model by Adcroft et al., (1997), as an attempt to
improve the representation of the bottom topography in general ocean circulation models. Adcroft et al.,
(1997) introduces partial bottom cells as a compromise solution between the less accurate but
computationally efficient full cell approach and the very accurate but computationally expensive shaved cell
approach. Partial bottom cells are implemented in FESOM2.0 by using the vertical ALE approach of
FESOM2.0 numerical core documented in Danilov et al. 2017.
Another feature made available through using ALE in FESOM2.0 is related to the sea ice-ocean interaction.
Naturally, sea ice, more precisely the loading of sea ice, contributes to the ocean pressure. However in many
ocean models, especially in the absence of surface mass fluxes or on fixed vertical grids, the loading is
omitted and sea ice is treated as "levitating". The option to consider sea ice loading is now implemented into
FESOM2.0, which is called "embedded" sea ice and was first introduced by Campin et al. (2008). They state
that the advection of sea ice in combination with the coupling of "embedded" sea ice through ice loading can
be an important source of ocean variability especially in the vicinity of ice edges (Campin et al. 2008). The
implementation of embedded sea ice relies on the zstar vertical-coordinate option in FESOM2 and also on
the fact that the sea ice component is called on each time step of the ocean model.
Diapycnal mixing in the ocean is an essential process that acts on the ocean stratification and the distribution
of heat, salt as well as passive tracers like nutrients, biological agents or $CO_2$. Various processes contributing
to diapycnal mixing can act with different magnitudes over a wide range of horizontal and vertical scales,
from several kilometers down to centimeters (Robertson and Dong, 2019). Due to the finite discretisation
scale in all ocean models, the mixing processes can not be resolved and thus must be parameterized. The
parameterisations of diapycnal mixing can be done in a variety of ways with different complexity, such as
boundary layer schemes like the K-profile parameterisation of Large et al. (1994) or turbulent closure
schemes like the one of Gaspar et al. 1990 and many others. A great innovation in the ocean modelling
community is the development of software packages that contain a variety of vertical mixing
parameterisations in a format that makes it easy to integrate them into existing model code (Fox Kemper et
al. 2019). One of these software packages is the Community Ocean Vertical Mixing package (CVMIX,
Griffis et al. 2015, Van Roekel et al. 2018), which now also was integrated into FESOM2.0. CVMIX is
tailored to be used in state of the art climate models to produce vertical profiles of diffusivity and viscosity
(Fox Kemper et al. 2019), providing a comparable mixing implementation over a wide spread of different
ocean models such as MOM6, POP, MPAS and ICON. Such effort makes it easier to compare these models





to each other. From the CVMIX package we implemented the Pacanowski and Philander 1981, the K-profile parameterization of Large et al. 1994 and the tidal mixing parameterisation of Simmons et al. 2004. Further, the infrastructure of the CVMIX library has been used to implement the turbulent kinetic energy (TKE) scheme of Gaspar et al. (1990) and the scheme for Internal Wave Dissipation and Mixing (IDEMIX) of Olbers and Eden (2013) in the same way as it is done in Gutjahr et al. (2020). It should be mentioned that neither TKE nor IDEMIX is yet part of the official CVMIX package but will hopefully be added to the package in the future.

Beside the prime vertical mixing schemes, like the K-profile scheme, the Pacanowski and Phillander scheme and others that have the purpose to deliver a usable mixing parameterisation for the entire ocean, and vertical mixing schemes like the tidal mixing scheme of Simmons et al. 2004 or IDEMIX that are used to parametrize internal wave processes which then result in a heterogeneous background diffusivity, there are also mixing parameterizations that aim at resolving regional processes. One of them was proposed by Timmerman and Beckmann (2004). It parameterises the wind driven mixing in the Southern Ocean especially when there is insufficient mixing during the melt seasons when other mixing schemes are used. It is used in FESOM2.0 to improve the otherwise too low stratification in the Southern Ocean and Weddell Sea.

The intention of this paper is to document the performance of the newly implemented features -- partial bottom cells, "embedded" sea ice, the vertical mixing parameterisations that come with the implementation of CMVIX and the local mixing parameterization of Timmerman and Beckmann (2004), based on comparing the associated hydrographic biases, changes in vertical convection and differences in Meridional Overturning Circulation, using a relatively coarse reference mesh.

The paper is structured as follows. First in Section 2 we describe the mesh configuration and model setup used in the simulations. The description and analysis of partial bottom cells, "embedded" sea ice and vertical mixing schemes is done in Section 3. A discussion and conclusion is given in Section 4.

## 2    Model configurations

We use here the FESOM2.0 coarse mesh configuration core2, which is the same mesh as in part 1. It consists of ~0.13M surface vertices, with a nominal resolution of 1° in the bulk of the ocean, ~25km north of 50°N, 1/3° in the equatorial belt and slightly enhanced resolution in the coastal regions. In the vertical, 48 unevenly distributed layers are used, with a vertical grid spacing stepwise increasing from 5m at the surface to 250 m towards the bottom.

All model simulations are initialised from the Polar Science Center Hydrographic winter Climatology (PHC3.0, updated from Steele et al., 2001) and forced by the CORE interannually varying atmospheric forcing fields (Large and Yeager, 2009) for the period 1948-2009. For each simulation a spin-up over three full CORE cycles was applied, where each subsequent cycle was initialised with the final results from the preceding cycle. All modelled data shown in this work are averaged over the period 1989-2009.




All model simulations except the one with the Turbulent-Kinetic-Energy (TKE) closure mixing of Gaspar et
al., 1990, use a non-constant latitude-dependent vertical background diffusivity with values between 10-4
m2/s and 10-6 m2/s, as described in Scholz et al., 2019. Further, all simulations use the Monin-Obukhov
length dependent vertical mixing parameterization of Timmermann and Beckmann, 2004 in the surface
boundary layer south of -50°S. The effects of this parameterisation on the simulated ocean state in
FESOM2.0 is described in section 3.4. The horizontal viscosity is computed via a modified harmonic Leith
approach (Fox-Kemper and Menemenlis, 2008) plus a biharmonic background viscosity (0.01 m²/s) . For
coarse-mesh setups, like the one used here, FESOM2.0 uses the Gent-McWilliams (GM) parameterisation
for eddy stirring (Gent et al., 1995; Gent and Mcwilliams, 1990) and we follow the implementation after
Ferrari et al., 2010. The isoneutral tracer diffusion (Redi, 1982) coefficient equals to that of GM, same as in
Scholz et al. (2019) and in previous FESOM versions (Wang et al. 2014). GM and Redi are scaled with
horizontal resolution with a maximum value of 3000 m²/s at 100 km horizontal resolution and change
linearly to zero between a resolution of 40 km and 30 km. In the vertical, they are scaled according to Ferrari
et al., 2010 and Wang et al., 2014. The simulations use as default the K-profile parameterisation for vertical
mixing (KPP, Large et al., 1994), a linear free surface (Scholz et al., 2019), levitating sea ice and a full
bottom cell approaches, unless otherwise stated.

## 126 3 FESOM2.0 model components and evaluation

### 127 3.1 Partial bottom cells

The concept of partial cells, as an attempt to improve the bottom representation in general ocean circulation
models, which was first introduced for the finite volume approach by Adcroft et al., (1997). Although an
early version of partial cells was developed by Cox, (1977), and used by Semtner and Mintz, (1977) and
Maier-Reimer et al., (1993), it has never got officially released (Griffies et al., 2000). Adcroft et al.
(1997) presented three different cases. The first one is the conventional full cell approach, where the depth of
the ocean bottom is approximated with the nearest standard depth level of the vertical model discretization.
The second one is the partial cell approach in which the bottom level can take any intermediate depth within
the cell, thus capturing water columns more accurately. In these two cases, the bottom features a "stepped"
topography and the jump of the steps is smaller for the partial cell approach (Adcroft et al., 1997). The third
case introduced by Adcroft et al., (1997) is a shaved cell approach, which assumes a constant slope within
each bottom cell and gives the best approximation for a continuous bottom topography. Adcroft et al.
(1997) showed that the shaved cell approach gives the most accurate results, but induces a significant
increase in computational demand, whereas the partial cell approach is a good compromise between the low
computational demand of the full cell approach and the increased accuracy of the shaved cell approach.
Hence, most ocean models (e.g. NEMO, MOM6, MPAS, POP) including FESOM2.0 went in favor of the





partial cell approach.
For the implementation of partial cells in FESOM2.0 we follow the work of Pacanowski and Gnanadesikan,
(1998), which implemented partial cells for the B-grid discretization in MOM2 with efforts to minimize
pressure gradient errors and spurious diapycnal mixing. They addressed that calculating horizontal pressure
gradients needs some special attention for partial cells since not all grid points within the bottom layer are at
the same depth. In FESOM2.0, we compute pressure gradient force based on the density Jacobian approach
as used by Shchepetkin, (2003) and not the pressure Jacobian approach proposed by Pacanowski and
Gnanadesikan, (1998). The density Jacobian approach is less prone to pressure-gradient error than using
pressure Jacobian, and therefore the model is more stable. Furthermore, we limited the thickness of the
partial bottom cell to be at least half of the full cell layer thickness to reduce the possibility of violating the
vertical Courant–Friedrichs–Lewy (CFL) criterion.
Using a B-grid like discretisation, where the scalars are located at vertices of a triangular mesh while the
velocities are located at the centroids of the triangular elements, makes it necessary to define the partial cells
at both locations. First, the partial bottom depth is defined at the centroids of the triangular elements based
on the real bottom topography considering the aforementioned limitation. Then, the vertice partial bottom
depth is derived from the deepest partial bottom of the surrounding triangular elements.
In order to demonstrate the effect of the partial cells on the simulated ocean state we performed two model
simulations using the full cell and partial cell approaches, respectively. We investigate, first, the temperature
biases of the full cell approach with respect to the data of the World Ocean Atlas 2018 (WOA18, Locarnini
et al., 2018; Zweng et al., 2018, in the left column of Fig. 1) and, second, the temperature differences
between partial cell and full cell (partial-full) averaged over five different depth ranges 0-250m, 250-500m,
500-1000m, 1000-2000m and 2000-4000m (in the right column of Fig. 1).
The full cell setup (Fig. 1, left) shows positive climatological temperature biasin the northern and southern
Pacific, the Atlantic equatorial ocean as well as in the central Indian Ocean through the depth ranges of 0-
250m, 250-500m and 500-1000m. In the same depth ranges there are also negative biases in the North
Atlantic (NA) subtropical gyre and in the equatorial and southern subtropical Pacific. The depth ranges of
250-500m and 500-1000m indicate cold biases in the Southern Ocean (SO) and around the coast of
Antarctica. The deeper depth ranges (1000-2000m and 2000-4000m.) indicate small negative temperature
biases in most of the world oceans, except for the Atlantic and Arctic Ocean (AO), which possess a small
warming bias in the depth ranges. The Arctic warming anomaly at these depths originates largely from a
vertically too much extended Atlantic water inflow branch (not shown), which is a typical feature of coarse
resolution models (e.g., Ilicak2016).
Using partial cells (Fig. 1, right) leads to profound changes especially at the position of zonal fronts in the
North and South Atlantic. In the depth ranges of 0-250 m, 250-500 m and 500-1000 m in the NA, partial
cells lead to a cooling in the Labrador Sea (LS) and Irminger Sea (IS) as well as along the path of the Gulf
Stream (GS) and North Atlantic Current (NAC), except for the area around -30°W, 50°N which is





characterised by warming. In the upper South Atlantic (SA), partial cells lead to a northward shift of Brazil–
Malvinas Confluence Zone expressed by a dipole of warmer South Atlantic Current (SAC) and cooler
Antarctic Circumpolar Current (ACC). Further, partial cells lead to a predominant cooling in the SO Atlantic
sector and parts of the Indian Ocean sector, while the Pacific sector of the SO and most of the Antarctic
coastal areas are dominated mostly by warming anomalies. The Arctic Ocean features a slight warming
anomaly at all depths, except for the surface, when using partial cells instead of full cells.
Fig. 2 shows the same as Fig. 1 but for salinity. Here, with respect to WOA18, the full cell run indicates a
generally fresher AO for the surface- and the 250-500 m depth range. Further negative salinity biases can be
found within the upper three depth ranges in the equatorial Pacific, north and south subtropical Atlantic, at
the position of the Atlantic northwest corner, northern IO as well as parts of the SO. Strong positive salinity
biases with full cells can be found in the surface depth range of the North Pacific and in the Chukchi- and
Beaufort Sea. Further positive salinity biases in the 250-500 m and 500-1000 m depth ranges are found along
the pathway of the Gulf Stream as well as in the equatorial Atlantic and central IO. The deep depth range of
1000-2000m has positive salinity anomalies in the Northern and Southern Atlantic and negative salinity
biases in the Mediterranean outflow branch and IO.
Using partial cells leads to an increase in salinity throughout all depth ranges of the AO relative to using full
cells. Further, a salinity increase at the position of the "cold blob", in the GIN sea, in the eastern South
Atlantic and parts of the SO can be observed within the upper three depth ranges. Compared to full cells,
using partial cells reduces salinity along the pathway of the GS, the Antarctic Circumpolar Current (ACC) in
the South Atlantic and along the coast of Antarctica.
The differences in the horizontal velocity speed between partial and full cells (Fig. 3), for the depth ranges of
0-250 m, 250-500 m, 500-1000 m, 1000-2000 m, 2000-4000 m and at the bottom, reveal that with partial
cells the velocity in the East Greenland Current (EGC), West Greenland Current (WGC ) and Labrador
Current (LC) are stronger by up to 0.02 m/s through all depth ranges presented here. The upper differences
reveal that partial cells lead to a weakening and a slight southwards shift of the NAC between -45°W and -
30°W, and a more pronounced tendency towards a northwest bend of the NAC between -30°W and -15°W,
which is nevertheless still too far eastward. By using partial cells the pathway of the Irminger Current (IC)
moves closer to the continental slope.
In terms of absolute and anomalous northern and southern hemispheric maximum mixed layer depth (MLD),
using partial cells leads to a slight MLD decrease in the southern LS, IS and northern Greenland-Iceland-
Norwegian (GIN) Seas, and a slight MLD increase along the pathway of the IC and in the southern and
central GIN Seas (Fig. 4c). In the southern hemisphere, partial cells have a more pronounced effect, leading
to a significant, up to 1000 m, decrease in MLD in the central Weddell Sea (WS) and a minor increase in
MLD of around 300 m along the eastern continental slope of the Antarctic Peninsula.
The differences between using full cells and partial cells in Global-, Atlantic- and Indo-Pacific Overturning
Circulation (Fig. 5) are rather small with magnitudes of less than 1Sv. Both cases feature an upper AMOC





circulation cell of ~16 Sv and an Antarctic Bottom Water (AABW) cell with strength between -1 Sv and -2 Sv. One can summarize that partial cells lead to a clear improvement of the circulation pattern, especially regarding the branch of the Gulf Stream and NAC even in rather coarse resolved configurations.

## 3.2    Embedded sea ice

As described in Scholz et al. (2019), FESOM2.0 supports the full free surface formulation with two possible options, zlevel and zstar (Adcroft and Campin, 2004). Both options allow for surface freshwater exchanges which can modify the thickness of the surface layer and thus decrease or increase salinity in the surface layer. This avoids the need of virtual salinity fluxes, which are required in the linear free surface (linfs) approach when the layer thicknesses are kept fixed. Using virtual salinity fluxes has the potential to affect the model integrity on long timescales and change local salinities with certain biases (Scholz et al., 2019).

In reality part of sea ice is embedded in the ocean with impact on the ocean pressure below. In the model, when the sea ice loading is omitted, the "levitating" sea ice (Campin et al., 2008) does not impose pressure on the ocean. This is the default case in the case of linfs but also applicable to zlevel and zstar. The other case when ice-loading is considered has "embedded" sea ice (Rousset et al., 2015), which depresses the sea surface according to its mass. Since it affects the layer thicknesses, this case is only available for the full free surface cases of zlevel and zstar. Although freezing and melting have no direct effect on the oceanic pressure, the divergence of the ice transport does modify the ice-loading fields and influences the hydrostatic pressure (Campin et al., 2008). As mentioned by Campin et al., 2008, this effect could be compensated by the divergence of the oceanic transport in the special case where sea ice and ocean velocities match, but in reality sea ice and ocean velocities are rarely identical especially in the presence of high frequency wind forcing. Therefore, sea ice dynamics in combination with the ice-loading coupling can be a source of oceanic variability especially near the ice-edge where ice divergence/convergence is large (Campin et al., 2008). However, using embedded sea ice harbours the risk that the amount of sea ice loading due to excessive accumulation and the resulting depression in the surface elevation may result in a depletion of the surface layer thickness, when the zlevel option is used, where only the surface layer is allowed to change. To avoid this issue, we limit in FESOM2.0 the maximum ice loading to a sea ice height of 5m when the zlevel option is used. In case of using zstar, the problem is less severe, since here the change in elevation is distributed over all vertical layers, except for the bottom one. This makes zstar to be the recommended option when using embedded sea ice, as also stated by Campin et al., 2008.

To show the effect of embedded sea ice on the simulated ocean state, two simulations were carried out using the zstar option of FESOM2.0, one with levitating (omitting the effect of sea ice loading on ocean pressure) the other with embedded sea ice (including the effect of sea ice loading on ocean pressure).

Fig. 6 shows the sea ice concentration (SIC) for March and September in the levitating sea ice case and the difference between the embedded and levitating sea ice cases. Superimposed are the simulated (solid) and observed (dashed, Cavalieri et al., 1996) contour line of the 15% sea ice extent. The northern hemispheric





March sea ice edge (Fig. 6a) shows a good agreement with observational data for the LS, IS and Bering Sea
but reveals a too far southwards extension in the Greenland Sea and Barents Sea. The simulated northern
hemispheric September sea ice extent (Fig. 6b) is larger than the observations. The southern hemispheric
March sea ice extent is underestimated in the simulation, while the simulated southern hemispheric
September sea ice extent is in good agreement with the observation.
Using the embedded sea ice leads to an increase in the SIC in the Greenland Sea by around 6% in March. In
September, embedded sea ice leads to positive SIC anomalies in the eastern- and negative anomalies in the
western AO. In the southern hemisphere, embedded sea ice leads to a heterogeneous pattern of small positive
and negative changes along the sea ice edge.
Regarding the changes in the ocean, Fig. 7 shows the temperature (left column) and salinity (right column)
differences between the embedded and levitating (embedded minus levitating) sea ice cases averaged over
the depth ranges 0-250 m, 250-500 m, 500-1000 m, 1000-2000 m and 2000-4000 m. The temperature and
salinity differences reveal that a significant warming of up to 0.5°C and a salinification of up to 0.10 psu
occurs in almost the entire AO due to embedded sea ice, except in a thin stripe along the eastern continental
shelf of the AO that shows negative anomalies in the depth ranges of 0-250 m, 250-500 m and 500-1000 m.
The changes in temperature and salinity can be explained by the changes in ocean currents. Figure 8 depicts
the speed of the horizontal currents in levitating (1$^{st}$ column) and embedded (2$^{nd}$ column) sea ice cases as
well as their difference (3$^{rd}$ column). Using embedded sea ice leads to an increase in the speed along the
entire boundary current of the Eurasian Basin and along the Lomonosov Ridge, that can be found in all three
presented depth ranges. The increase in the velocity of the boundary currents, caused by using embedded sea
ice, leads to an enhanced heat and salt transport in the Atlantic water layer originating from the Fram Strait,
which results in a warmer and more saline intermediate depth in the Arctic Ocean. The increase in
temperature and salinity, especially in the surface layers of the AO using embedded sea ice reduces existing
local biases (see Fig. 1 and Fig. 2) that occur when using levitating sea ice. On the whole it can be stated that
using embedded sea ice instead of levitating sea ice has some significant effect on the ocean dynamics of the
AO, but no effect in the Southern Ocean or Antarctic marginal seas.

## 3.3 Implementation and evaluation of vertical mixing schemes

Besides the already existing Pacanowski and Philander (fesom_PP, Pacanowski and Philander, 1981) and
MOM5 K-profile (fesom_KPP, Large et al., 1994) vertical mixing parameterizations in FESOM2.0 that were
based on the implementation in the predecessor version FESOM1.4, the vertical mixing parameterizations of
the Community Vertical Mixing (CVMIX, Griffies et al., 2015) project have been now added as well. This
includes the CVMIX vertical mixing of: Pacanowski and Philander (cvmix_PP), the MOM6 K-profile
(cvmix_KPP) parameterization, the tidal mixing parameterization of Simmons et al., (2004) (cvmix_TIDAL)
and the turbulent kinetic energy (cvmix_TKE) mixing of (Gaspar et al., 1990) in combination with the
Internal Wave Dissipation, Energy and Mixing (IDEMIX) parameterization (Olbers and Eden, 2013 and





Eden and Olbers, 2014) . Although cvmix_TKE and IDEMIX are not yet a part of the CVMIX project, they
use its libraries in the background and will join the project in the future. CVMIX is used by a variety of
models, such as MOM6, POP, MPAS or ICON and provides an opportunity of a cross model-spanning
vertical mixing implementation that allows for an enhanced cross-model intercomparison.

**290  3.3.1  Comparison of cvmix_KPP, cvmix_PP with previous fesom_KPP and fesom_PP**
**291  implementation**

In FESOM2.0 we implemented cvmix_PP and cvmix_KPP in addition to its previous implementations
fesom_PP and fesom_KPP that were adopted from FESOM1.4. The difference between cvmix_PP and
fesom_PP lies in the background coefficient for viscosity which is considered in cvmix_PP but not in
fesom_PP when computing the diffusivity, following the experience with FESOM1.4 which did not need to
be more diffusive. The difference between cvmix_KPP and fesom_KPP lies mainly in the treatment of the
squared velocity shear and buoyancy difference with respect to the surface. In cvmix_KPP the surface
quantities are computed by averaging over the Monin-Obukov surface layer (Griffies et al. 2015) while in
fesom_KPP the surface values are linked to the first layer in the model.
Suppl. 2 displays the temperature (1st and 2nd column) and salinity (3rd and 4th column) biases of fesom_KPP
with respect to WOA18 (1st and 3rd column) as well as the difference between fesom_PP and fesom_KPP (2nd
and 4th column). In the surface depth range the climatological temperature and salinity biases of fesom_KPP
with respect to WOA18 are largely negative in the tropical and subtropical Pacific, North and South Atlantic
as well as AO, and positive in tropical Atlantic and Indian Ocean, Southern Ocean, Labrador Sea, GIN Seas
and the marginal seas of the North Pacific. The subsurface depth ranges of 250-500 m and 500-1000 m are
dominated by largely positive temperature biases, except for the Southern Ocean, the pathway of the GS and
NAC and the northern Indian Ocean. The salinity biases in the 250-500 m and 500-1000 m depth range
preserve largely the pattern from the surface layer except for an increasing and expanding positive salinity
bias in the tropical Atlantic, reduced positive salinity biases in the Indian Ocean and northern Pacific as well
as reduced negative biases in the Arctic Ocean. The 1000-2000 m depth range features small warm biases in
the AO and GIN seas, positive temperature and salinity biases in the LS and the South Atlantic, negative
temperature and salinity biases in the eastern North Atlantic (possibly due to weak mediterranean outflow)
and small negative temperature and salinity biases in the Pacific and Indian Ocean. The very deep depth
range of 2000-4000 m reveals rather small warming bias for the entire Atlantic and SO.
Fesom_KPP and fesom_PP produced rather small temperature and salinity differences (note different
colorbar ranges between 1st & 2nd and 3rd & 4th column), considering the biases with respect to the WOA18
climatology. Employing fesom_PP has the tendency to be slightly warmer almost everywhere in the
subsurface layers and slightly saltier especially in the AO and fresher in the surface layer of the subtropical
and equatorial ocean compared to using fesom_KPP.
Fig. 9 shows the difference in temperature (1st column), salinity (2nd column) and vertical diffusivity (3rd





column) between cvmix_KPP and fesom_KPP (cvmix_KPP minus fesom_KPP) averaged over five different depth ranges. The last column presents the fesom_KPP vertical diffusivity as a reference. Also here, the temperature and salinity differences are rather small compared to the climatological biases shown in Suppl. 2. cvmix_KPP has the tendency to produce in the marginal seas of the AO a slightly fresher surface ocean, while the central AO shows an increase in salinity by ~0.1 psu.

The absolute value of the vertical diffusivity in fesom_KPP is larger than that in cvmix_KPP in the surface layers as well as in regions of unstable stratification (buoyancy frequency < 0), superimposed on a non-constant background diffusivity as described in Scholz et al., 2019. The different treatment of the squared velocity shear and buoyancy difference with respect to the surface in cvmix_KPP leads to a reduction of the vertical diffusivity ($3^{rd}$ column) in the Labrador and Irminger Seas and to an increase in the AO locally by up to one order of magnitude (especially in the deep ocean).

The differences in MLD between fesom_KPP and cvmix_KPP are presented in Fig. 10, where and a) and b) show the absolute MLD value for fesom_KPP in the northern hemisphere in March and in the southern hemisphere in September respectively. Fig. 10 c) and d) display the corresponding anomalies between cvmix_KPP and fesom_KPP (cvmix_KPP-fesom_KPP). The absolute MLD values for fesom_KPP in March show high values of up to 3300 m in the entire LS and parts of the Irminger Sea, intermediate values of up to 2000 m in the northern and eastern GIN seas and values of ~900m along the eastern continental slope of the North Atlantic. In the southern hemisphere in September, fesom_KPP simulates large MLD of ~2500 m in the central Weddell Sea and weaker MLD of ~500 m in the band of the Antarctic Circumpolar Current (ACC). Compared to the fesom_KPP, cvmix_KPP leads to a ~200 m weaker MLD in the boundary currents of the LS, southern LS and along the northeastern continental slope of the GIN seas, and slightly larger MLD values in the IS and southwestern GIN Seas.

Fig. 11 presents the differences in temperature ($1^{st}$ column), salinity ($2^{nd}$ column) and vertical diffusivity Kv ($3^{rd}$ column) between cvmix_PP and fesom_PP (cvmix_PP minus fesom_PP) as well as the absolute values of vertical diffusivity for fesom_PP ($4^{th}$ column). For the upper two surface depth ranges, cvmix_PP shows an overall small warming anomaly, except for the Gulf of Guinea in the 250-500 m depth range where the anomaly is negative. The salinity with cvmix_PP has overall slight positive anomalies, except for coastal Arctic areas and the Gulf of Guinea which indicate a slight freshening anomaly when compared to fesom_PP. The depth ranges below 500 m show no significant temperature or salinity differences between cvmix_PP and fesom_PP. The absolute value of Kv in fesom_PP shows also larger values all over the surface layer as well as in the areas of unstable stratification similar to fesom_KPP, but with a lower magnitude and a more extended region of increased Kv in the LS and IS. The Kv difference between cvmix_PP and fesom_PP shows sporadically positive values along the coastal Arctic Ocean and in parts of the North Atlantic and GIN Seas. As one would expect, cvmix_PP has an order of magnitude larger values in the very deep ocean layer where the background viscosity enters the computation of Kv in cvmix_PP.

Fig. 12 presents the absolute and anomalous MLD between fesom_PP and cvmix_PP. The MLD in





fesom_PP in March is deep in the entire LS and in parts of the IS, but slightly weaker and less spatially
extended when compared to fesom_KPP (Fig. 10). The MLD in the GIN seas is very similar between
fesom_PP and fesom_KPP. In the southern hemisphere the September MLD in fesom_PP shows a pattern in
the central Weddell Sea which is similar to that in fesom_KPP, but shallower by ~500 m. The MLD
difference between cvmix_PP and fesom_PP in the northern hemisphere indicates a very heterogeneous
pattern for the North Atlantic and in the southern hemisphere an up to ~150 m deeper MLD in the Weddell
Sea MLD for cvmix_PP compared to fesom_PP. Overall, the difference in the simulation results induced by
the difference in the two implementations of mixing schemes is generally small when considering the model
biases relative to observations.

### 3.3.2 Effects of tidal mixing parameterization of Simmons et al. (2004)

The tidal mixing parameterization of Simmons et al., (2004) provided by CVMIX has been added to
FESOM2.0. This mixing parameterization takes into account effects from internal wave generation due to
tides over rough bottom topography. The breaking of internal waves in the vicinity of topographic features
excites small-scale turbulence and leads to an enhanced vertical mixing. The tidal mixing parameterization
uses a two dimensional map of tidal energy dissipation flux due to bottom drag and energy conversion into
internal waves from Jayne and St. Laurent, (2001). It is transformed under consideration of a vertical
redistribution function, the modelled buoyancy frequency and a tidal dissipation efficiency and mixing
efficiency into a 3D map of diapycnal tidal vertical mixing, which is added to a primary vertical mixing
scheme like PP, KPP or TKE. To show the effect of the tidal mixing parameterization we conducted a
simulation using both cvmix_KPP and the tidal vertical mixing (cvmix_KPP$_{TIDAL}$). This simulation will be
compared with a control run with cvmix_KPP in which the tidal mixing is not considered. The differences in
temperature (1$^{st}$ column), salinity (2$^{nd}$ column) and vertical diffusivity Kv (3$^{rd}$ column) between
cvmix_KPP$_{TIDAL}$ and cvmix_KPP averaged over five different depth ranges are presented in Fig. 13. The last
column of Fig. 13 shows the cvmix_KPP Kv as a reference. The temperature anomalies of the upper three
depth ranges indicate that cvmix_KPP$_{TIDAL}$ is colder especially in the marginal seas of the North Pacific, e.g.
Sea of Japan, Sea of Okhotsk and Bering Sea, within the branch of the Gulf Stream (GS) and North Atlantic
Current (NAC) as well as in the GIN- and Barents Seas. The Arctic Ocean shows a cooling anomaly for the
500-1000 m and 1000-2000 m depth range. In the southern hemisphere the entire Southern Ocean is slightly
colder when including the tidal vertical mixing. The tropical and subtropical ocean indicates a slight
warming for cvmix_KPP$_{TIDAL}$.
The salinity anomalies between cvmix_KPP$_{TIDAL}$ and cvmix_KPP show a pattern similar to that of the
temperature, with a freshening in the marginal seas of the North Pacific, GS, NAC, GIN- and Barents Seas as
well as for the Southern Ocean. The upper depth range indicates an increase in salinity for the AO, while the
subsurface depth ranges show an AO freshening when including the tidal mixing. The tropical and
subtropical ocean shows largely an increase in salinity under cvmix_KPP$_{TIDAL}$.
The difference in vertical diffusivity shows for cvmix_KPP$_{TIDAL}$ an increase by an order of magnitude along





all topographic features which is induced by the tidal vertical mixing parameterization. On top of that the central AO shows a reduced vertical diffusivity by at least an order of magnitude for the 250-500 m, 500-1000 m and 1000-2000 m depth ranges, which comes from a change in local hydrography when including the tidal vertical mixing parameterization and the associated difference in the KPP mixing scheme.

To further understand the effect of the tidal vertical mixing, Fig. 14 shows the global zonal mean temperature and salinity differences between the case of cvmix_KPP ando the WOA18 (a, c) and the differences between cvmix_KPP$_{TIDAL}$ and cvmix_KPP (b, d). The temperature of cvmix_KPP shows a rather strong warming bias until 1000 m for the tropical and subtropical ocean as well as until ~2500 m for the ocean north of 50°N with respect to WOA18 (Fig. 14a). The deep ocean features small negative temperature anomalies for the tropical and subtropical ocean and slightly positive biases for the deep SO, when compared to WOA18. The salinity biases of the cvmix_KPP case (Fig. 14c) indicate a more heterogeneous but nevertheless similar picture. Also here positive salinity biases can be seen in the tropical and subtropical ocean until around 1000m as well as until ~2500m for the ocean north of 50°N. Looking at the temperature and salinity difference between cvmix_KPP$_{TIDAL}$ and cvmix_KPP, it can be seen that the tidal mixing of Simmons et al., (2004) leads to a cooling and freshening of the Southern Ocean and the ocean north of 50°N as well as a warming and salinification for the tropical and subtropical ocean until around 1500m. The deep ocean experiences a general slight warming and freshening due to the inclusion of the tidal mixing parameterization. In general one can summarize that the tidal mixing parameterization of Simmons et al., (2004) helps to improve some of the biases with respect to WOA18. The last panel in Fig 14e shows the global zonal averaged vertical diffusivity profiles between cvmix_KPP$_{TIDAL}$ and cvmix_KPP and reveals a general strong increase in Kv along the continental slope in the southern ocean, in the northern hemisphere north of 50°N as well as in the deep ocean interior.

To illustrate the effect of Simmons et al., (2004) tidal mixing parameterization onto the MLD, Fig. 15 presents the northern hemisphere March (a) and southern hemisphere September (b) MLD in the case of cvmix_KPP, and the difference in MLD between cvmix_KPP$_{TIDAL}$ and cvmix_KPP also for northern hemisphere March (c) and southern hemisphere September (d). In the northern hemisphere in March, tidal mixing leads to an increase in the MLD within the boundary currents of the LS, southern and eastern GIN Seas as well as in the Sea of Okhotsk. In the southern hemisphere September, tidal mixing leads to a significant ~1000 m increase in the Weddell Sea MLD. This significant increase originates largely from enhanced mixing of very cold surface waters along the continental slope of the Weddell Sea due to the tidal mixing parameterization.

### 3.3.3 Effects of Turbulent-Kinetic-Energy (TKE) mixing parameterisation

More elaborate parameterizations of the vertical mixing in the ocean can be achieved by using closure schemes of turbulent kinetic energy (TKE) and the associated turbulent mixing within the mixed layer and below. One of these turbulent closure schemes is by Gaspar et al. (1990) that has been implemented via CVMIX (cvmix_TKE) into FESOM2.0 based on the work of Eden et al. (2014) and Gutjahr et al. (2020).





The turbulence closure scheme requires the solving of the second-order equation for TKE which is closed by connecting the vertical diffusivity with the turbulent kinetic energy and a length scale for its dissipation (Eden et al., 2014). For the background diffusivity we do not use here the latitude and depth dependent background diffusivity as in the previous mixing schemes. Instead, a constant minimum value of TKE is assumed, which takes into account the ocean interior mixing by internal wave breaking. To understand the effect of cvmix_TKE on oceanic hydrography, Fig. 16 presents the temperature and salinity biases of cvmix_TKE with respect to WOA18 (1st and 3rd column). To relate cvmix_TKE to the other vertical mixing schemes (e.g. KPP), the temperature and salinity differences between fesom_KPP and cvmix_TKE (2nd and 4th column) are shown as well. In general, the cvmix_TKE temperature and salinity biases with respect to WOA18 look largely very similar to the biases of fesom_KPP shown in Supp2. 1 (1st and 3rd column) in terms of the spatial patterns. A closer inspection of temperature and salinity differences between cvmix_TKE and fesom_KPP (Fig. 18, 2nd and 4th column) reveals that cvmix_TKE produces an up to 0.5°C colder ocean within the 0-250 m, 250-500 m and 500-1000 m depth ranges in most of the ocean, a strong warming along the pathway of the NAC and the southern polar front in the South Atlantic, and small warming biases in the AO and SO. The salinity differences between cvmix_TKE and fesom_KPP indicate a salinification of the AO throughout the 0-250 m, 250-500 m and 500-1000 m depth ranges, but most pronounced in the surface depth range. The surface saline bias largely stems from reduced mixing under sea ice, which shields the ocean from the wind stress, a large source term of TKE. Furthermore, there are positive salinity anomalies in the North Atlantic (in the pathway of the GS and NAC), North Pacific and Southern Ocean, and largely negative salinity anomalies in the southern hemisphere. The temperature and salinity differences between cvmix_TKE and fesom_KPP in the depth ranges of 1000-2000 m and 2000-4000 m are rather marginal. It should be mentioned that a part of the anomalies described here could also be attributed to the different treatment of the background diffusivity. fesom_KPP takes a latitude and depth dependent value (Scholz et al., 2019), while cvmix_TKE assumes a constant value of minimum TKE on the surface (10e-4 $m^2/s^2$) and for the interior mixing (10e-6 $m^2/s^2$).

### 3.3.4 Effects of energy consistent combination of TKE with the Internal Wave Dissipation Energy and Mixing (IDEMIX) parameterisation

Besides the standard implementation of vertical background diffusivity in cvmix_TKE using a constant minimum value of TKE to parameterize the effect of breaking of internal waves, cvmix_TKE also allows for the usage of a more sophisticated parameterization of internal wave breaking when combined with the IDEMIX parameterization (Olbers and Eden, 2013; Eden et al., 2014) which describes the energy transfer from sources towards sinks of internal waves by using a radiative transfer equation of weakly interacting internal waves. The resulting dissipation of energy is then treated as a source term in the turbulent kinetic energy balance equation leading at the end to an energetically more consistent interpretation of the internal ocean mixing process (Eden et al., 2014; Gutjahr et al., 2020). Thereby, IDEMIX solves for the propagation of low-mode internal waves far from their generation sites, which is considered by Fox-Kemper et al., (2019)





as one of the most difficult components of the internal wave energy budget. Different from the tidal mixing parameterization of Simmons et al., (2004), which only represents the generation of internal waves by barotropic tides and their breaking at rough topography, IDEMIX considers both the internal waves due to barotropic tides and the internal waves induced by wind-stress fluctuations and exiting at the base of the mixed layer (Gutjahr et al., 2020). The combination of cvmix_TKE and IDEMIX to improve the energetic consistency of ocean models is a rather new approach in the modelling community. It has been evaluated for stand-alone ocean models (Eden et al., 2014; Nielsen et al., 2018; Pollmann et al., 2017) and coupled models (Nielsen et al., 2019). Further, the computed TKE dissipations rates from IDEMIX have been evaluated against observational Argo float-derived dissipation rates by Pollmann et al. (2017) and have been found to be in good agreement (Gutjahr et al., 2019). In this part of the FESOM2 documentation, two FESOM2.0 simulations with cvmix_TKE, one with and one without the usage of IDEMIX, are compared to assess the effect of IDEMIX on the modelled hydrography.

Fig. 17 presents the temperature (1$^{st}$ column), salinity (2$^{nd}$ column) and vertical diffusivity (3$^{rd}$ column) differences between cvmix_TKE with IDEMIX versus without it, averaged over five different depth layer ranges. As a reference the vertical diffusivity of cvmix_TKE without IDEMIX is also shown in the 4$^{th}$ column. The temperature differences indicate a clear warming of all equatorial and mid-latitudinal oceans and a cooling in the AO, SO and the marginal seas of the North Pacific throughout almost all the depth ranges, when cvmix_TKE is used with IDEMIX. There is a particularly strong warming in the surface and subsurface depth range of the North Atlantic, in the subsurface depth range of the south Pacific and in the deeper depth ranges of the Indian Ocean. The salinity differences (2$^{nd}$ column) have a similar spatial pattern, showing a rather strong salinification of the equatorial and mid-latitudinal global oceans and a freshening of the AO, SO and North Pacific from the surface to 500-1000 m depth range. The depth ranges below indicate a predominant general freshening almost everywhere, except for the Mediterranean outflow and Indian Ocean which indicate a slight warming. The differences in the vertical diffusivity between cvmix_TKE with and without IDEMIX are only very small in the upper layer depth range. Therefore, all subsurface depth layers indicate considerable positive vertical diffusivity differences by up to two orders of magnitude especially along all major topographic features as well as in the SO. This shows in particular how IDEMIX parameterizes the vertical mixing due to the breaking of upwards propagating internal wave excited by barotropic tides along the ocean bottom topography but also the vertical mixing related to the internal wave breaking of downward propagating internal waves radiated out of the mixed layer like e.g. in the SO.

Fig. 18 presents the global zonal mean temperature and salinity differences of cvmix_TKE with respect to the WOA18 (a, c) as well as the temperature, salinity and vertical diffusivity differences between cvmix_TKE$_{IDEMIX}$ and cvmix_TKE (b, d, e). The zonal mean temperature biases of cvmix_TKE with respect to WOA18 (Fig. 18a) are positive for the upper SO, the equatorial and mid-latitudinal oceans between 500m until 1000m, and the high-latitude ocean north of 60°N where the warming bias extends nearly from the surface until a depth of ~2500m. A rather weak warming bias is also present for the very deep >2500m SO.





General cooling biases can be seen for the equatorial and mid-latitudinal surface oceans, between a depth of ~1000m to 2000m as well as for the very deep ocean. The salinity biases for cvmix_TKE (Fig. 18c) show too high salinities for the high-latitude ocean north of 40°N and for the surface SO. Small salinity biases can be found in the equatorial and mid-latitudinal surface layers as well as around 40°N between ~1000 and 3000 m.

The temperature differences between cvmix_TKE with and without IDEMIX (Fig. 18b) shows that the IDEMIX leads to a general warming of the equatorial and mid-latitudinal oceans especially between ~500 m and ~2000 m, but a cooling in the northern and southern high-latitude oceans. The salinity differences between cvmix_TKE with and without IDEMIX reveal a similar pattern with an increase in salinity for the equatorial and mid-latitudinal ocean from the surface until a depth ~2000m and a freshening bias in the same depth range for the high-latitudinal oceans and for the entire deep ocean as well.

The corresponding vertical diffusivity difference is shown in Fig. 18e. There, using IDEMIX results in an increase in vertical diffusivity along the bottom topographic slopes in the SO and north of 50°N until 70°N. Further, an increase in vertical diffusivity can be observed for almost the entire upper ocean until ~2000 m with deeper reaching positive anomalies between -60°S - 30°S and 30°N - 50°N. A reduction of the vertical diffusivity can be observed for the entire AO from the surface to bottom, for the equatorial and mid-latitudinal deep ocean >3000 m as well as for the deep (>4000 m) SO.

The effect of IDEMIX on the MLD is presented in Fig. 19, which shows the northern hemisphere March a) and southern hemisphere September b) cvmix_TKE MLD and the corresponding anomalies between cvmix_TKE with and without IDEMIX. It indicates that the use of IDEMIX leads to an increase in northern hemisphere MLD within the boundary currents of the LS by up to ~1000 m and in the southeastern GIN Seas by up to ~1800 m. In the southern hemisphere September, IDEMIX leads to a significant increase of the Weddell Sea MLD up to ~1800 m. We observe that using cvmix_KPP$_{TIDAL}$ or cvmix_TKE$_{IDEMIX}$ the model cannot maintain the upper halocline in the Weddell Sea. Hence the warm water that shall stay deep is exposed to the surface and the ocean loses heat. It can be well seen from Fig. 14.b and 18.b as blobs of negative temperature differences beneath the surface. As a consequence, the enlarged MLDs in the Weddell Sea appear. We therefore recommend to combine cvmix_KPP$_{TIDAL}$ or cvmix_TKE$_{IDEMIX}$ with the partial bottom cell approach, which has a compensating effect on the stratification in the Weddell Sea (see section 3.1 and Suppl. 1) due to improvements of the current circulation in the Weddell Sea.

### 3.4 Implementation of Monin-Obukhov length dependent vertical mixing

In this section the effect of the Monin-Obukhov length vertical mixing (MOMIX) of Timmermann and Beckmann (2004) in FESOM2.0 is discussed. In an attempt to decrease the climatological biases especially in the Southern Ocean, which were otherwise prone to significant cooling and salinification (not shown), MOMIX has been implemented into FESOM2.0 as well. MOMIX serves as a parameterisation of the wind driven mixing in the Southern Ocean, effective especially in the melting season, which helps to reduce





winter deep convection in the Weddell Sea, thus affecting the basin wide ocean- and meridional overturning
circulation (Timmermann and Beckmann, 2004). MOMIX computes the Monin-Obukhov length based on
heat flux, freshwater flux, wind stress, sea ice concentration and sea ice velocity following the approach of
Lemke (1987), and subsequently increases the vertical diffusivity within the Monin-Obukhov length to a
value of 0.01m²/s.
Due to its success in reducing the aforementioned mean biases, MOMIX is applied at the moment in
FESOM2.0 per default south of -50°S. In the following, the effects of MOMIX are discussed, based on
simulation of fesom_KPP and cvmix_TKE each with and without MOMIX.
Fig. 20 presents the temperature (1st and 2nd column) and salinity (3rd and 4th column) differences between
simulations with and without MOMIX for both the fesom_KPP and cvmix_TKE schemes, averaged over
five different depth ranges. Using MOMIX in the Southern Ocean leads to a significant warming of up to
1°C for almost the entire Southern Ocean south of -60°S throughout all considered depth ranges, except for
the surface depth range of the southern Weddell Sea and subsurface southern Pacific which exhibits cooling
anomalies. The warming anomaly is slightly more pronounced for fesom_KPP than cvmix_TKE. The usage
of MOMIX in the Southern Ocean leads in fesom_KPP to a warming of the Gulf Stream and to a cooling of
the NAC. For cvmix_TKE this behaviour is reversed. The salinity anomalies indicate a freshening for the
entire Southern Ocean surface depth range when using MOMIX, while the subsurface depth ranges indicate
predominantly a slight increase in salinity, except for the southern Weddell Sea 250-500m depth range.
To emphasize the effect of MOMIX on the Weddell Sea MLD, Fig. 21 presents the Southern Ocean
September MLD for fesom_KPP (a) and cvmix_TIDAL (b) without MOMIX and the corresponding
anomalies with minus without MOMIX (c, d). The MLD for fesom_KPP (a) and cvmix_TKE (b) are very
large over the entire Weddell Sea and parts of the Ross Sea. The MLD values are higher and more extended
with fesom_KPP than with cvmix_TKE. However, for both vertical mixing schemes without using MOMIX,
the MLD values are way too high within the Weddell Sea and Ross Sea. The figures c) and d) visualize what
happens with the Southern Ocean MLD for fesom_KPP and cvmix_TKE when MOMIX is used. Especially
for fesom_KPP, MOMIX leads to a significant decrease in the MLD in almost the entire Weddell Sea of up
to ~3000 m, except for the southwestern Weddell Sea close to the continental shelf which exhibits an
increase in MLD. Also the large MLD patch in the Ross Sea becomes strongly reduced when using MOMIX.
Both fesom_KPP and cvmix_TKE face the same pattern in MLD reduction when using MOMIX, only the
magnitude in the MLD decrease is larger in fesom_KPP than in cvmix_TKE.
Since MOMIX has a rather strong effect in reducing the Weddell Sea open-ocean deep-water formation it
will also consequently affect the formation of Antarctic Bottom Water (AABW) and the Meridional
Overturning Circulation (MOC). Fig. 22 shows the fesom_KPP global (a), Atlantic (b) and Pacific (c) MOC
when MOMIX is switched off and the difference from the case with MOMIX (bottom row). It can be seen
that on a global but also basin-wide scale, the use of MOMIX leads to a reduction in the strength of the
AABW, in the Atlantic by ~0.6 Sv and in the Pacific by up to ~1.7 Sv. Also the strength of the upper AMOC



cell is reduced by ~1 Sv when using MOMIX. We conclude that using MOMIX helps to alleviate the
problem of large MLDs in the Weddell Sea which we addressed above. Hence, the options cvmix_KPP$_{TiDAL}$
or cvmix_TKE$_{IDEMIX}$ are strongly recommended to be used in combination with MOMIX, which is per
default active only South of -50°S.

## 577   4      Discussion and Conclusions

This paper describes the two new features -- partial cells and embedded sea ice introduced to FESOM2.0 and
the implementation of the vertical mixing library CVMIX (cvmix_PP, cvmix_KPP, cvmix_TKE, IDEMIX
and cvmix_TIDAL), together with the elaboration of the effect of MOMIX. These new features expand the
functionality of FESOM2.0, its applicability and its ability to be better compared to other state of the art
ocean general circulation models. With its model components implemented, FESOM2.0 is mature for its
practical applications and holds its leading role in the competition of the global unstructured ocean models.
We demonstrate the effect of using partial cells by comparing them against the full cell approach. It is shown
that partial cells lead to an improved representation of the Gulf Stream branch, with a reduction in the cold
bias in the northwest corner of the North Atlantic associated with an improved NAC pathway. Further,
partial cells lead to a "northwest corner like" meridional deflection of the NAC between -30°W and -15°W
which is still too far east, but leads to an improved representation in a rather coarse configuration which
would otherwise be dominated by a rather zonal NAC. Partial cells also lead to a general speed up of the
boundary currents shown as an example for the North Atlantic.
The improvement of the NAC pathway and the speedup of the boundary currents especially in the subpolar
gyre by using partial cells is described by a variety of publications (e.g. Barnier et al., 2006; Käse et al.,
2001; Myers, 2002). Besides all its advantages, partial cells also harbor the risk of increasing the existing
biases, like in our coarse configuration the deep Arctic warm bias, which is largely inherited from the
Atlantic Water inflow branch that expands too deep. The tendency of partial cells to increase the velocity in
the boundary currents leads to an enhancement of the Atlantic Water inflow to the Arctic Ocean. As the
temperature in the Arctic Atlantic Water layer is already overestimated without using partial cells, the warm
bias becomes even larger when partial cells are used. However, this is not the principle drawback of partial
cells, but rather an issue of model tuning for the pan-Arctic region, which is part of our on-going work (for
example, evaluating different numerical schemes of momentum viscosity). In the southern hemisphere, using
partial cells leads to a significant reduction of the otherwise rather high MLD in the Weddell Sea. Regarding
the configuration used in this paper, using partial cells leads to a strengthening of the warm deep water
current (Vernet et al. 2019) that crosses the Weddell Sea interior. Thus it enhances the local stratification
(see Suppl. 1 white arrow) and reduces vertical convection. It can be summarized that the usage of partial
cells clearly improves the general circulation within FESOM2.0 and that the benefits outweigh the
drawbacks.




The second feature that was presented, is the effect of embedded sea ice vs. the standard case of levitating sea ice. Embedded sea ice allows for a further step towards a more realistic and physical ocean-sea ice interaction by adding the sea ice loading to the ocean pressure. This has the potential of increasing ocean variability especially near the sea ice edge. Our results indicate that the embedded sea ice has only a minor effect on the sea ice distribution itself. Nevertheless the effect is the strongest for the Northern Hemisphere summer, when the sea ice edge retracts towards the Arctic Ocean interior. Here embedded sea ice leads to an up to 9% increase in the sea ice concentration in the eastern Arctic Ocean marginal seas, which also leads to an increase in the bias of the sea ice edge, and to a 6% decrease in the marginal seas of the western Arctic Ocean, which slightly reduces the sea ice extent bias there. The effect of embedded sea ice on the hydrography of the Arctic Ocean is much more significant, with an increase in temperature and salinity of up to 0.5°C and 0.1psu, respectively through most of the upper 1000 m. The increase in temperature and salinity is connected to a particular increase of the boundary currents especially along the eastern boundaries of the Eurasian Basin but also to a strengthening of the cyclonic current along the Lomonosov Ridge, which was otherwise rather weakly represented in the levitating sea ice case. The deficiencies of the Arctic Ocean currents representation in our model configuration can be partially attributed to the rather coarse resolution. However, with embedded sea ice we seem to be able to at least partly counteract the effect of low resolution and improve the Arctic Ocean current structure at rather low costs. We note that embedded sea ice could also deteriorate the model results in some cases. Since the boundary currents around the Eurasian Basin get enhanced, the already existing Atlantic Water layer biases get enhanced. However, as mentioned above, this is an issue of model tuning with this coarse resolution setup, not a drawback of embedded sea ice itself.

628

To further expand the functionality and comparability of FESOM2.0 we implemented the vertical mixing library CVMIX and its components, which in our implementation include cvmix_PP, cvmix_KPP, cvmix_TIDAL, cvmix_TKE and cvmix_TKE+IDEMIX. At first, the vertical mixing parameterizations fesom_KPP and fesom_PP, which have been already implemented in FESOM2.0, are briefly evaluated. It is shown that fesom_PP produces a slightly colder tropical and subtropical but warmer polar oceans on the surface, with a largely warmer ocean below the surface layer depth range, when compared to fesom_KPP. This makes between these two, fesom_KPP the preferred vertical mixing option at least in terms of mean temperature biases. In terms of salinity biases, fesom_PP performs better in the surface and subsurface AO as well as in the equatorial Atlantic and Indian Ocean, while otherwise fesom_KPP indicates smaller biases. In the next instance, fesom_KPP and cvmix_KPP have been compared to each other, since there are slight differences in their implementation. The difference in implementation leads only to minor differences in temperature throughout all considered depth ranges. Regarding the salinity differences, cvmix_KPP produces a considerably fresher surface AO compared to fesom_KPP, which is attributed to a reduced near surface vertical diffusivity in cvmix_KPP that leads to an over-stabilisation of the AO halocline. This enhances the





mean salinity bias in that region. In terms of vertical diffusivity, cvmix_KPP has the tendency to produce by up to one order of magnitude lower value (especially in the very deep depth range) in the main convection areas of Labrador Sea and Greenland Sea, throughout all considered depth ranges, accompanied by increased diffusivity in the subsurface of the Arctic Ocean. The reduced diffusivity in the main convection areas is attributed to the different treatment of the shear- and buoyancy difference with respect to the surface in cvmix_KPP that leads to a reduction of the local ocean boundary layer depth and to slightly reduced maximum MLD in Labrador and Greenland Sea, while the maximum MLD in the Weddell Sea becomes slightly enhanced, when using cvmix_KPP over fesom_KPP.

Since the implementation of cvmix_PP and fesom_PP are also slightly different, we also compare them. Although the produced diffusivities between cvmix_PP and fesom_PP are very similar, cvmix_PP indicates a further warming and salinification in the surface and 250-500 m depth ranges except for the upwelling region in the Gulf of Guinea which indicates a cooling and freshening and the surface depth range of the Arctic Ocean where it creates a predominant freshening, when compared the fesom_PP. The MLD values indicate that cvmix_PP leads in FEOSM2.0 to a slightly stronger convection in the Weddell Sea. The differences between fesom_PP and cvmix_PP are related to the different treatment of the background coefficient for viscosity when computing the diffusivity see Pacanowski and Philander (1981).

The effect of implementing cvmix_TIDAL in combination with cvmix_KPP was further assessed. cvmix_TIDAL serves here as a resourceful way to heterogenize the effect of tidally induced internal wave breaking that is otherwise homogenized in a constant or latitude dependent value for the background diffusivity. Using cvmix_TIDAL clearly leads to an enhancement of the vertical diffusivity along the slopes of the bottom topography, where tidally related internal wave breaking is induced. This leads especially in the high-latitude marginal seas, e.g. Sea of Okhotsk and Bering Sea but also Arctic Ocean and Southern Ocean, to a decrease in temperature and salinity due to the enhanced mixing along their shelfs. This enables cvmix_TIDAL to improve some of the existing local temperature and salinity biases within FESOM2.0 at rather low computational costs. However, the enhanced vertical diffusivity along the shelf of the Weddell Sea weakens the stratification and leads to a further increase in the MLD of the Weddell Sea of up to 1000 m.

Further, the implications of TKE vertical mixing parameterisation in FESOM2.0, added by Eden et al. (2014) and Gutjahr et al. (2020) to the CVMIX library, was evaluated based on a comparison with fesom_KPP. It is shown that the mean temperature and salinity differences between cvmix_TKE (Fig. 17) and fesom_KPP (Fig. 9) show very similar patterns. cvmix_TKE tends to produce a generally colder tropical and extratropical ocean together with slightly warmer polar oceans when compared to fesom_KPP. The salinity differences between cvmix_TKE and fesom_KPP shows that cvmix_TKE tends to produce a significantly saltier surface layer AO, revealing a much smaller salinity bias for the Arctic Ocean interior. This is largely connected to enhanced surface vertical mixing along the Arctic Ocean shelf break (not shown) within cvmix_TKE, that helps to partly destabilize the AO halocline. The improvement of the Arctic Ocean





hydrography when using cvmix_TKE is also found by Gutjahr et al. (2020) in the coupled ocean-atmosphere
Max Planck Institute Earth System Model (MPI-ESM1.2). Further, cvmix_TKE leads to a salinity increase in
the entire North Atlantic and northwest Pacific marginal seas, while the southern hemisphere, except for the
Southern Ocean, shows a freshening when compared to fesom_KPP. The reduced temperatures and salinities
in the tropics and extratropics when using cvmix_TKE are connected to the reduced vertical mixing.
However the regions of strong vertical shear, e.g. the branch of the Gulf Stream and NAC as well as
Southern Ocean show stronger vertical mixing in cvmix_TKE, when compared to fesom_KPP (not shown),
which is accompanied by positive temperature and salinity anomalies between cvmix_TKE and fesom_KPP.
Following the comparison of cvmix_TKE and fesom_KPP, a side by side comparison of cvmix_TKE with
and without IDEMIX was carried out. Here IDEMIX provides an alternative formulation of the background
diffusivity in cvmix_TKE using a radiative transfer equation of weakly interacting internal waves (Olbers
and Eden 2013), where energy is transferred from sources of internal waves to wave sinks, such as the
breaking of internal waves, which provide a source for TKE, leading to an energetically more consistent
treatment of internal mixing (Eden et al. 2014). As compared to the tidal background mixing
parameterization of Simmons et al (2004), IDEMIX allows not only for the generation of internal waves by
barotropic tides interacting with marine topography, but also for their propagation in the horizontal and
vertical directions away from region of generation and their damping due to wave-wave interaction or
interaction with the continental shelf. Further, IDEMIX allows for the excitation of internal waves at the base
of the mixed layer by high frequency wind forcing (Eden et al. 2014).
The combined TKE + IDEMIX approach was already applied in a couple of publications (Eden et al. 2014,
Nielsen et al. 2018, Gutjahr et al. 2020). It was shown in Pollmann et al. 2017 that TKE dissipation rates
from the combined TKE+IDEMIX approach are comparable to dissipation rates estimated from Argo floats.
In FESOM2.0, the usage of TKE+IDEMIX leads to a significant increase in the tropical and extratropical
and to a decrease in the high-latitude temperature and salinity over depth when compared to the case of only
using cvmix_TKE. These differences compensate for some of the biases in the surface and intermediate
depth ranges when IDEMIX is not used. The usage of IDEMIX leads to an enhanced heterogeneous
representation of vertical mixing especially below the mixed layer along the continental shelves and
topographic slopes. However the temperature gain for the deeper depth ranges below 1000 m seems to be
strongly overestimated when using cvmix_TKE+IDEMIX, hinting at a too strong vertical mixing in the deep
ocean. When it comes to the MLD, cvmix_TKE+IDEMIX leads in the northern hemisphere to a significant
increase in the MLD along the Labrador Sea boundary currents and in the southern GIN seas, which can be
attributed to the enhanced mixing along the continental slope of the North Atlantic and in the vicinity of the
overflow regions. In the southern hemisphere using IDEMIX leads to an enhancement of the vertical
diffusivity along the continental slope of the Weddell Sea. This leads to an enhanced mixing of cold and
salty waters, which further reduces the stratification and significantly increases the MLD of the Weddell Sea
and to a rather overestimation of the otherwise already high MLD values.



This is in contrast to the findings of Gutjahr et al. 2020, who found that in their coupled MPI-ESM1.2 simulation, IDEMIX led to a reduction of the vertical mixing in the Weddell Sea allowing for more local stratification. On possibility to overcome the lack of performance of IDEMIX but also of cvmix_TIDAL in the Southern Ocean and Weddell Sea could be its combination with partial bottom cells, which had the tendency to significantly reduce the deep convection in the Weddell Sea. At this point it needs further studies also with FESOM2.0 to analyse the different behaviour of IDEMIX that could be influenced by local resolution, coupled ocean-atmosphere feedback or just different background water mass structure. Nevertheless, the achievable energetic consistency with the combined cvmix_TKE+IDEMIX approach is an interesting feature that should find more applications in the ocean modelling community, although there is still some way to go to better understand and improve its integration.

The last part in this paper deals with the vertical mixing parameterisation MOMIX of Timmermann and Beckmann, (2004) in FESOM2.0 that helped us to overcome some major biases in the model. Since the very beginning of FESOM2.0 the model suffered from a severe cooling and salinification bias in the Southern Ocean and marginal seas around Antarctica, that was accompanied by a strongly overestimated MLD values and too weak stratification in the Weddell Sea. It is shown here that applying MOMIX south of -50°S helped to significantly reduce the biases and bring the MLD depth values in the Weddell Sea into a reasonable range. MOMIX increases the vertical diffusivity within the depth range of the Monin-Obukhov mixing length. This helps the warmer and fresher surface water masses from the melting season to connect with colder and saltier subsurface water masses from the freezing season and thus increase the stratification and reduce the vertical convection. The reason why FESOM2.0 in the Southern Ocean is so dependent on MOMIX, which was not the case with FESOM1.4, needs further research.

To summarize, this paper is the second part of the documentation of the development of important key components of FESOM2.0 in a realistic global model configuration. We described the implementation of partial cells and embedded sea ice and their impact on the modelled hydrography. Furthermore, we briefly described the already existing vertical mixing parameterisation of fesom_KPP and fesom_PP as well as the newly introduced mixing parameterization of cvmix_PP, cmix_KPP, cmix_TIDAL, cvmix_TKE and cvmix_TKE+IDEMIX that came with the incorporation of the vertical mixing library CVMIX into FESOM2.0.

# 5    Code availability

The FESOM2.0 version used to carry out the simulations reported here is available on zenodo through https://doi.org/10.5281/zenodo.4742242. The used mesh, as well as the temperature, salinity and vertical velocity (for the calculation of the MOC) data of all conducted simulations, can be found under https://swiftbrowser.dkrz.de/tcl_s/hituvPNH3xwiIy/FESOM2.0_evaluation_part2_scholz_etal.    Simulated results can of course also be obtained from the authors upon request. Mesh partitioning in FESOM2.0 is based on a METIS version 5.1.0 package developed at the Department of Computer Science and Engineering



43

at the University of Minnesota (http://glaros.dtc.umn.edu/gkhome/views/metis, last access: 18 November 2019). METIS and the pARMS solver (Li et al., 2003) present separate libraries which are freely available subject to their licenses. The Polar Science Center hydrographic climatology (Steele et al., 2001) used for model initialization and the CORE-II atmospheric forcing data (Large and Yeager, 2009) is freely available online. The vertical mixing library CVMIX is freely available from https://github.com/CVMix/CVMix-src or https://doi.org/10.5281/zenodo.1000801

## Author contributions

SD, DS, PS and NK worked on the development of the FESOM2.0 model code and the tuning of the model. All simulations shown in this paper were carried out by PS who were also responsible for preparing the basic manuscript. QW, SD, NK, DS and TJ have contributed to the final version of the manuscript.

## Acknowledgements

This paper is a contribution to the project S2: Improved parameterisations and numerics in climate models, S1: Diagnosis and Metrics in Climate Models and M5: Reducing spurious diapycnal mixing in ocean models of the Collaborative Research Centre TRR 181 "Energy Transfer in Atmosphere and Ocean" funded by the Deutsche Forschungsgemeinschaft (DFG, German Research Foundation) – project no. 274762653, and the Helmholtz initiative REKLIM (Regional Climate Change). This study has benefited from funding from the Initiative and Networking Fund of the Helmholtz Association through the project "Advanced Earth System Modelling Capacity (ESM)". Dmitry Sein was also supported in the framework of the state assignment of the Ministry of Science and Higher Education of Russia (№0128-2021-0014).

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



51

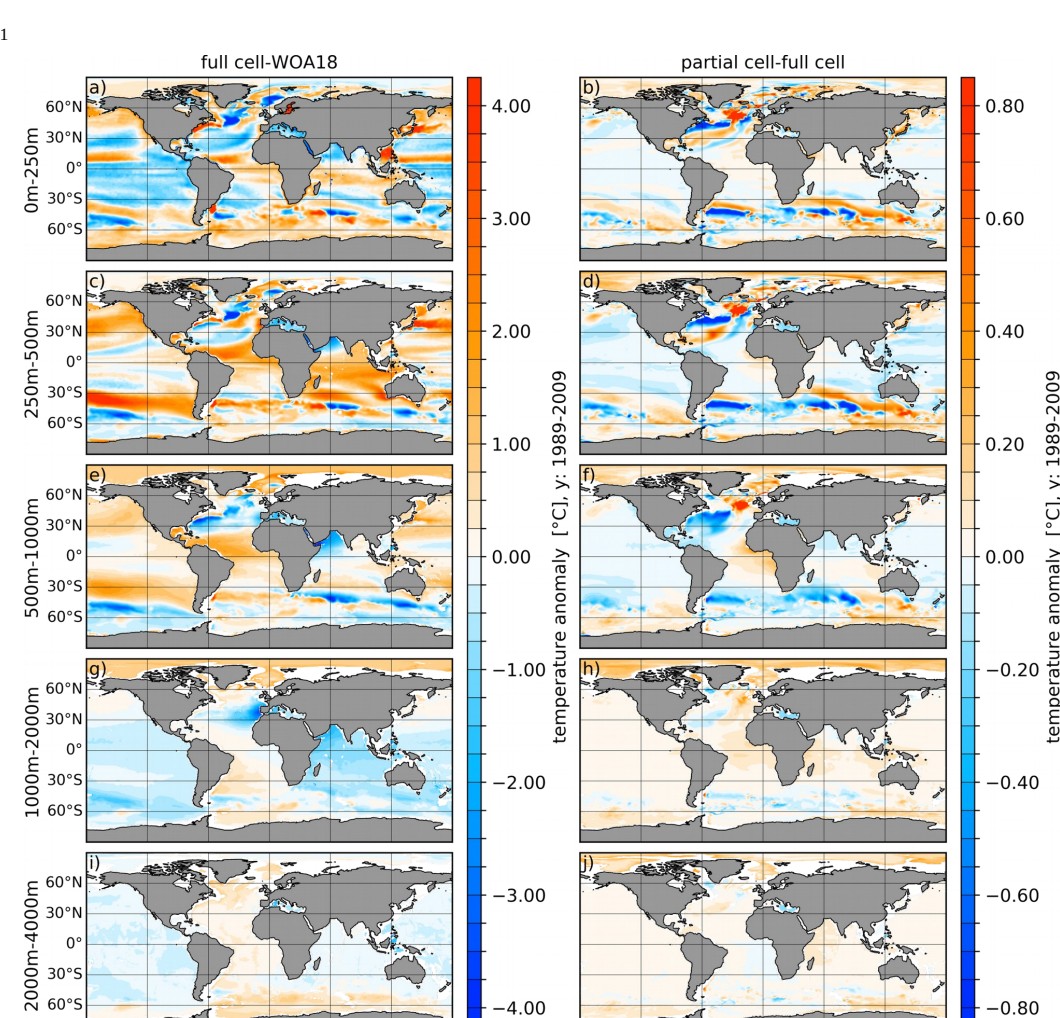

**Figure 1:** (Left column) Temperature biases full cells referenced to the World Ocean Atlas 2018 (WOA18, Zweng et. al 2018) averaged over the period 1989-2009. The right column shows the temperature difference between partial and full cells (partial minus full). From top to bottom the panels show the vertically averaged fields for the depth ranges of 0-250 m, 250-500 m, 500-1000 m, 1000-2000 m and 2000-4000 m.





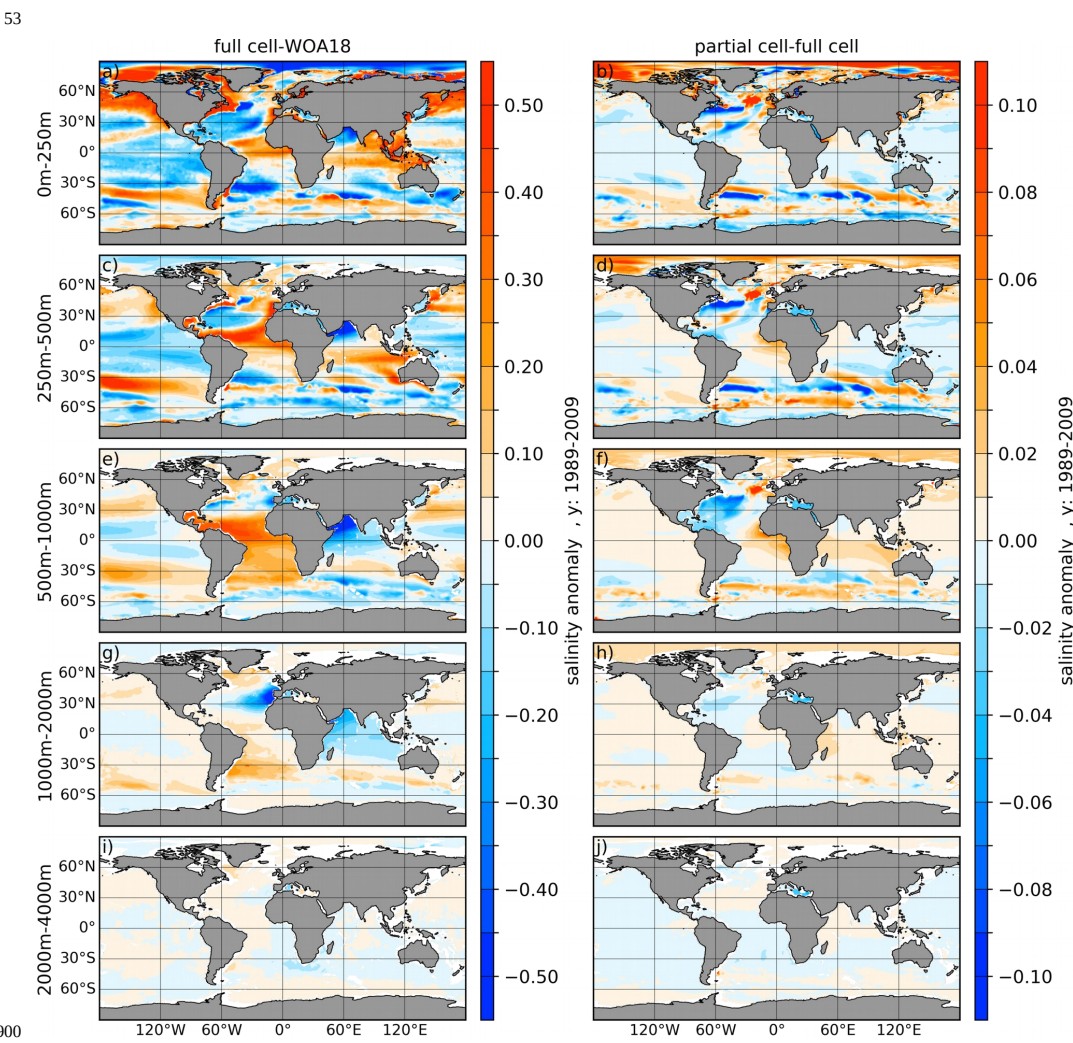

**Figure 2:** Same as Fig. 1, but for salinity.



55

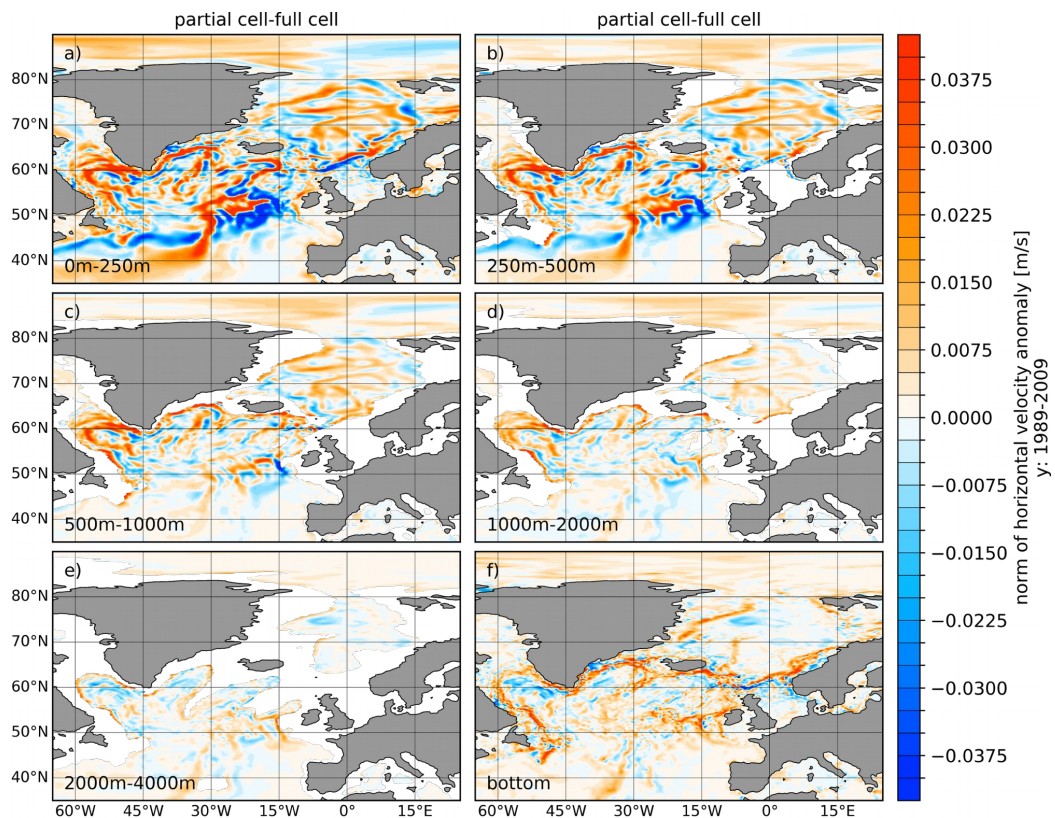

**Figure 3:** Difference of the horizontal velocity norm between simulations with partial and full cells (partial-full) averaged over the period 1989-2009 and averaged over the depth ranges of 0-250 m, 250-500 m, 500-1000 m, 1000-2000 m and 2000-4000 m as well as the bottom value.

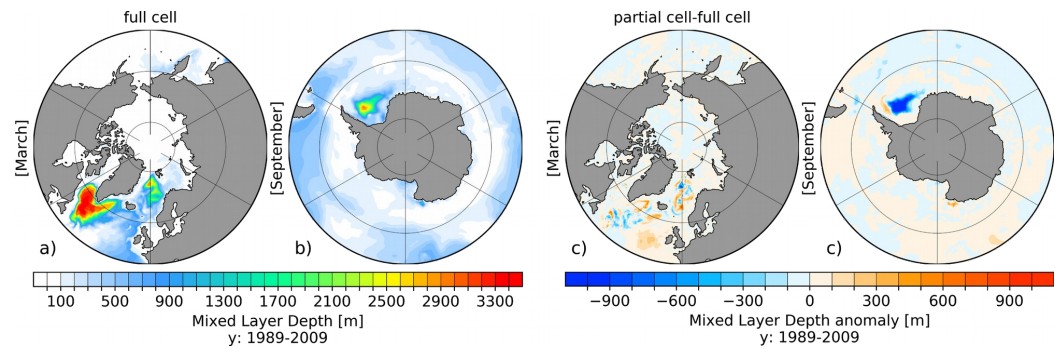

**Figure 4:** Northern hemispheric March (a) and southern Hemispheric September (b) mixed layer depth (MLD) with full cells as well as corresponding anomalous MLD with partial minus full cells (c, d), averaged for the period 1989-2009.





57

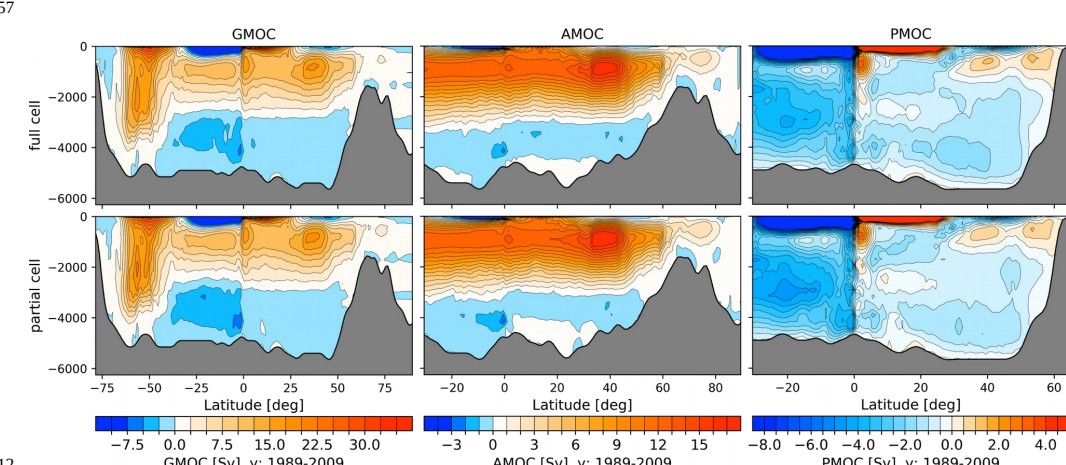

912

**Figure 5:** Global (GMOC, left column), Atlantic (AMOC, middle column) and Indo-Pacific (PMOC, right column) Meridional Overturning Circulation for full cell (upper row) and partial cell (lower row) averaged for the time period 1989-2009.

916

917



59

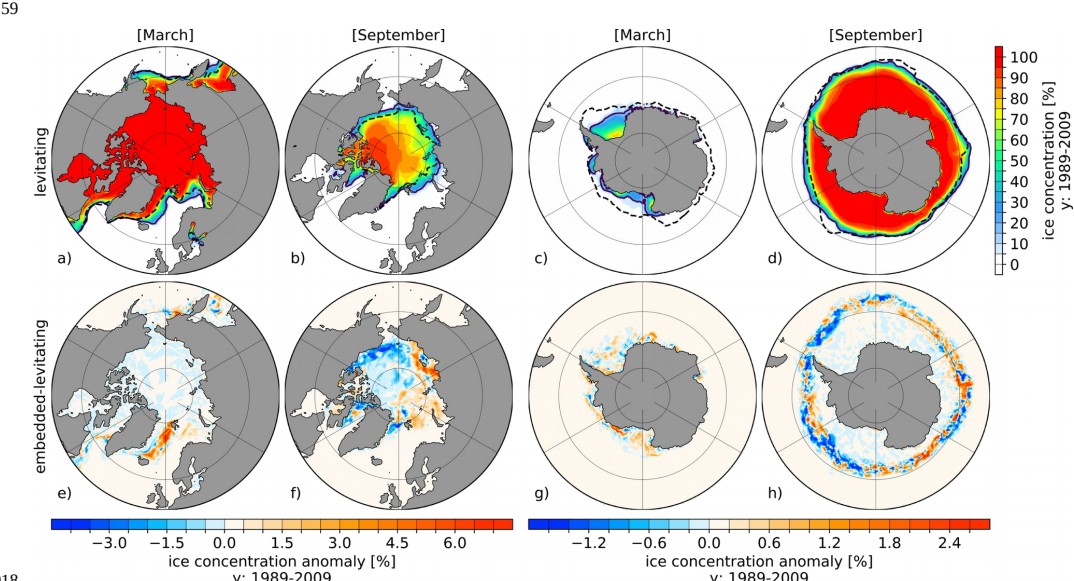

918

**Figure 6:** Levitating (upper row) northern and southern hemispheric March (a, c) and September (b, d) sea ice concentration averaged for the period 1989-2009. Solid and dashed lines indicate the simulated and observed (Cavalieri et al., 1996) contour of the 15% sea ice extent. The lower row shows the corresponding sea ice concentration anomalies between embedded and levitating sea ice (embedded minus levitating ) averaged over the same period.

924

61

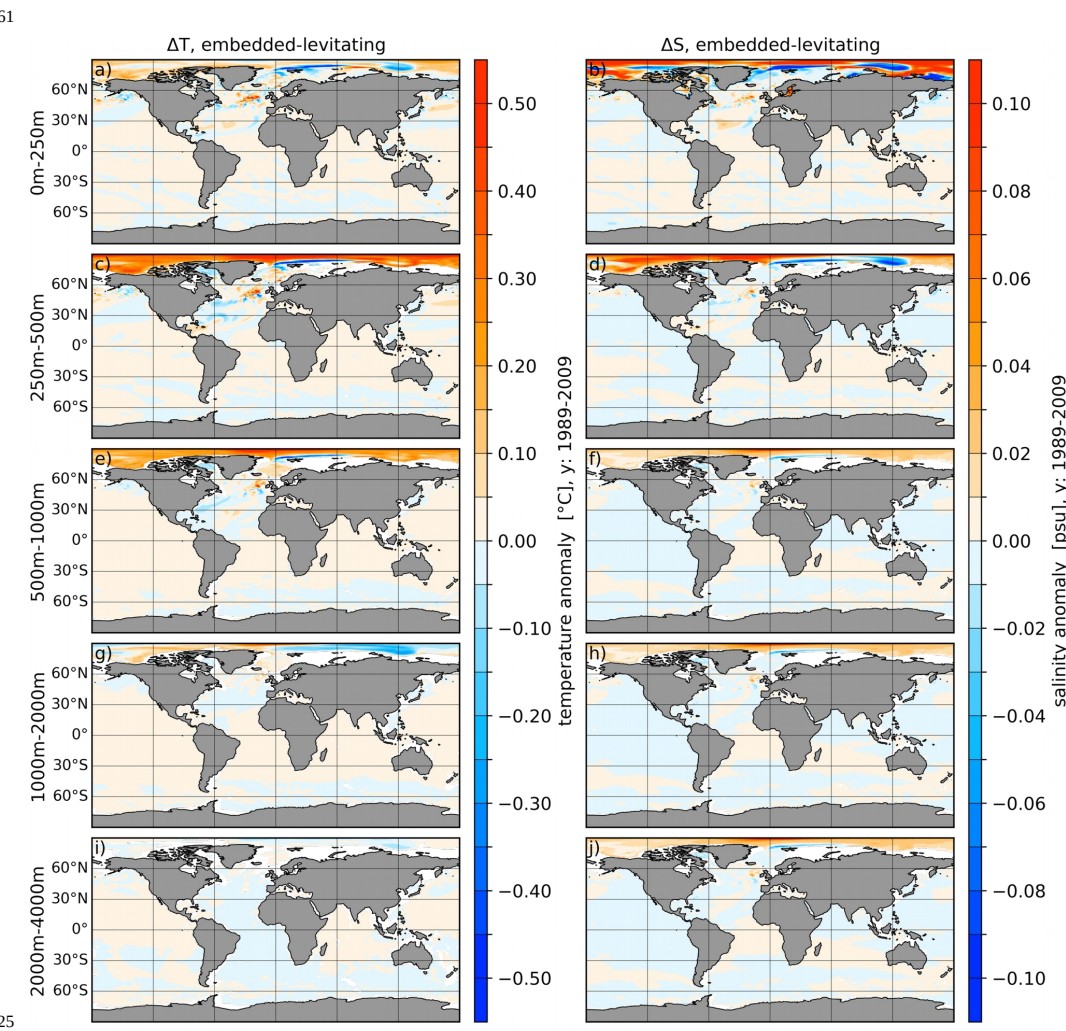

**Figure 7:** Temperature (left column) and salinity (right column) difference between embedded- and levitating sea ice averaged for the period 1989 to 2009. From top to bottom, panels show the vertically averaged fields for the depth ranges of 0-250 m, 250-500 m, 500-1000 m, 1000-2000 m and 2000-4000 m.






62



63



**Figure 8:** Norm of ocean velocity for levitating (left column) and floating (middle column) and the difference between embedded and levitating (right column) sea ice averaged for the period 1989 to 2009. From top to bottom, the panels show the vertically averaged fields for the depth ranges of 0-250 m, 250-500 m and 500-1000 m.



65

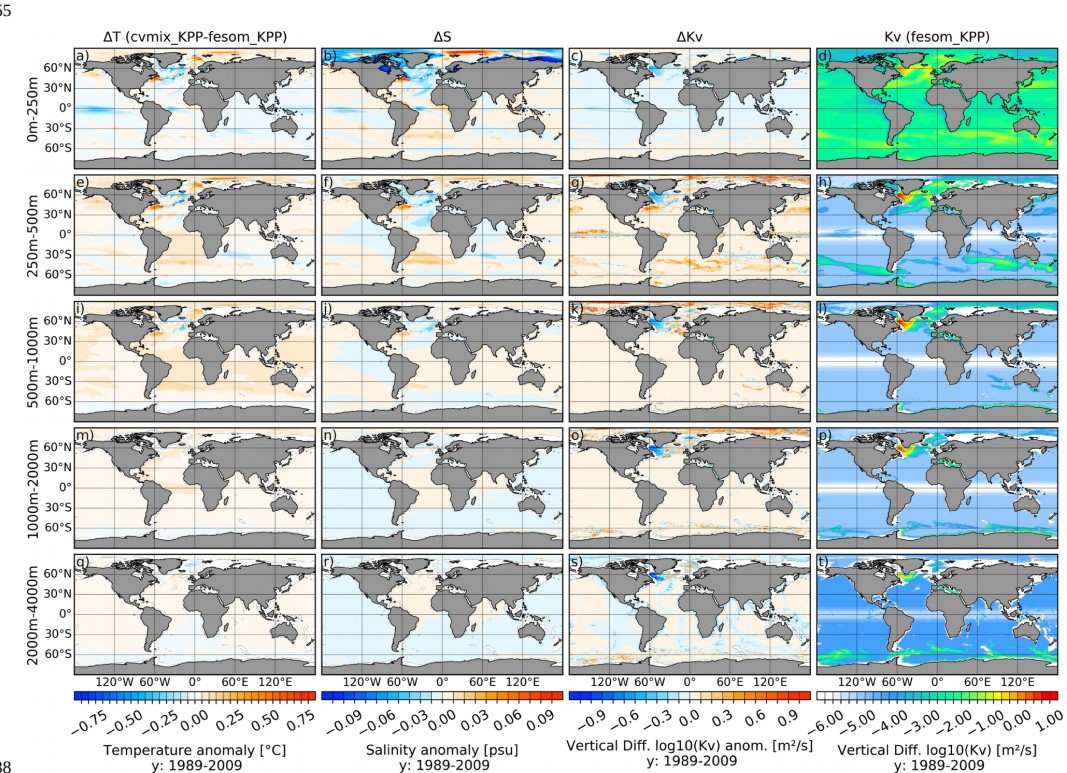

**Figure 9:** Temperature (1st Column), salinity (2nd column) and vertical diffusivity (3rd column) difference between cvmix_KPP and original fesom_KPP implementation as well as the absolute vertical diffusivity values (4th column) for fesom_KPP averaged for the period 1989 to 2009. From top to bottom, panels show the vertically averaged fields for the depth ranges of 0-250 m, 250-500 m, 500-1000 m, 1000-2000 m and 2000-4000 m.

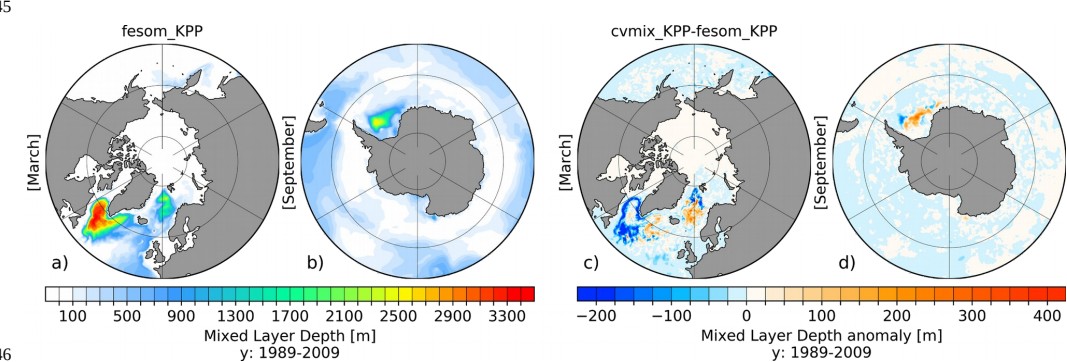

**Figure 10:** Northern hemispheric March (a) and southern Hemispheric September (b) mixed layer depth (MLD) for original FESOM2.0 KPP implementation as well as corresponding anomalous MLD between CVMIX and original FESOM2.0 KPP implementation (c, d), averaged for the period 1989-2009.



67

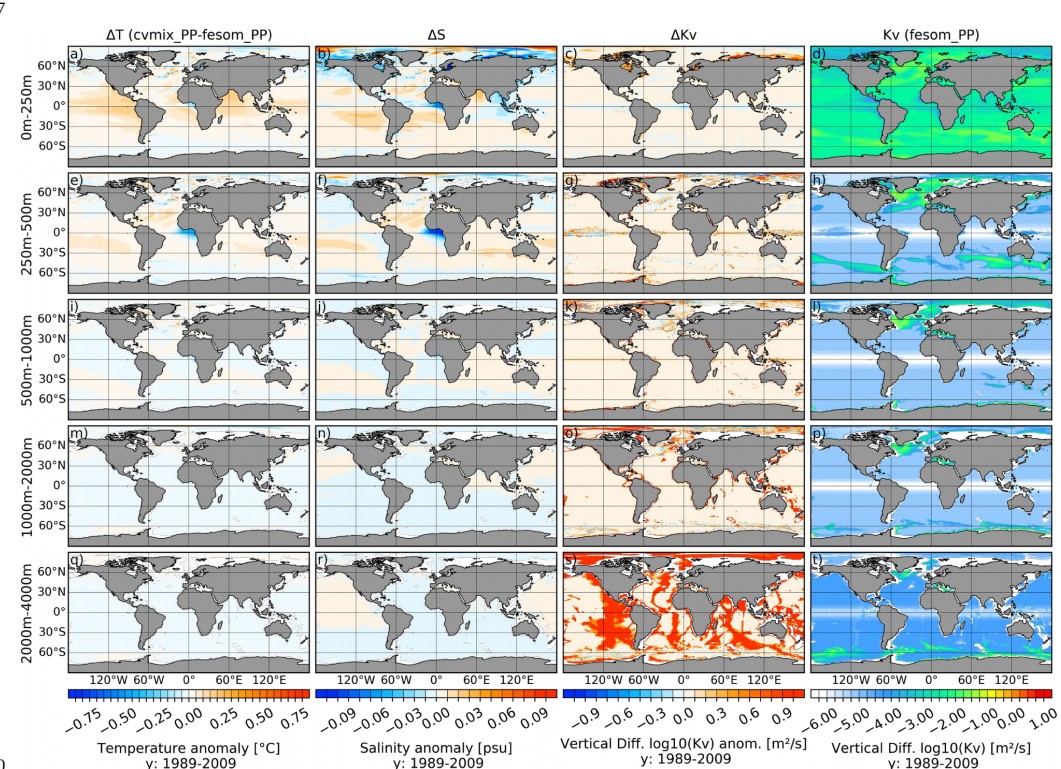

**Figure 11:** Temperature (1st Column), salinity (2nd column) and vertical diffusivity (3rd column) difference between cvmix_PP and original fesom_PP implementation as well as the absolute vertical diffusivity values (4th column) for fesom_PP averaged for the period 1989 to 2009. From top to bottom, panels show the vertically averaged fields for the depth ranges of 0-250 m, 250-500 m, 500-1000 m, 1000-2000 m and 2000-4000 m.

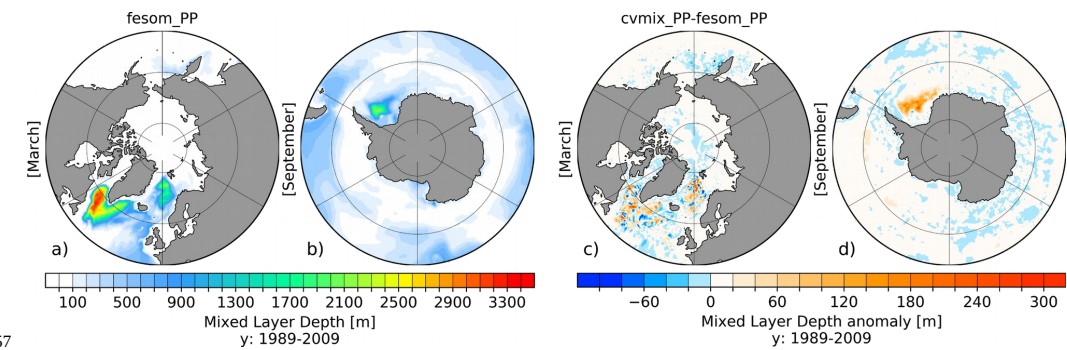

**Figure 12:** Northern hemispheric March (a) and southern Hemispheric September (b) mixed layer depth (MLD) for original FESOM2.0 PP implementation as well as corresponding anomalous MLD between CVMIX and original FESOM2.0 PP implementation (c, d), averaged for the period 1989-2009.

69

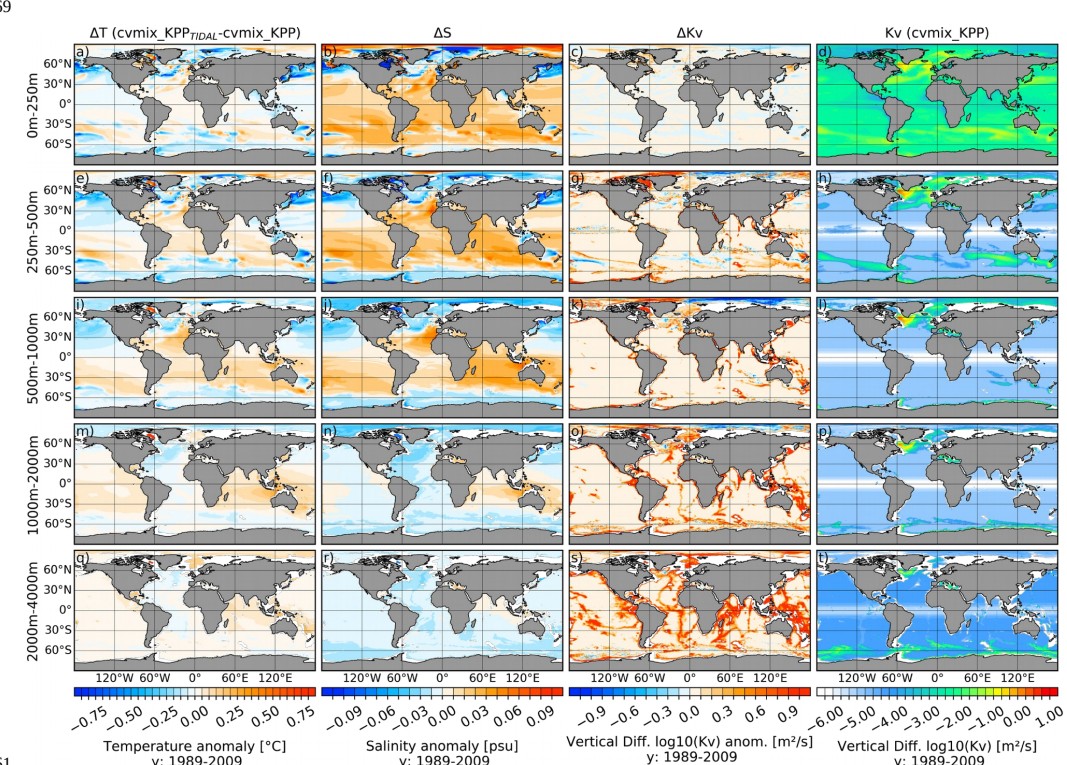

**Figure 13:** Temperature (1st Column), salinity (2nd column) and vertical diffusivity (3rd column) difference between CVMIX KPP with and without TIDAL mixing of Simmons et al. (2004) as well as the absolute vertical diffusivity values (4th column) for CVMIX KPP without TIDAL mixing averaged for the period 1989 to 2009. From top to bottom, panels show the vertically averaged fields for the depth ranges of 0-250 m, 250-500 m, 500-1000 m, 1000-2000 m and 2000-4000 m.

**Figure 14:** Global zonal averaged biases of temperature (a, b), salinity (c, d) and vertical diffusivity (e) profiles of cvmix_KPP with respect to WOA18 (a, c) and of cvmix_KPP with tidal mixing of Simmons et al. (2004) versus without (c, d, e).

73

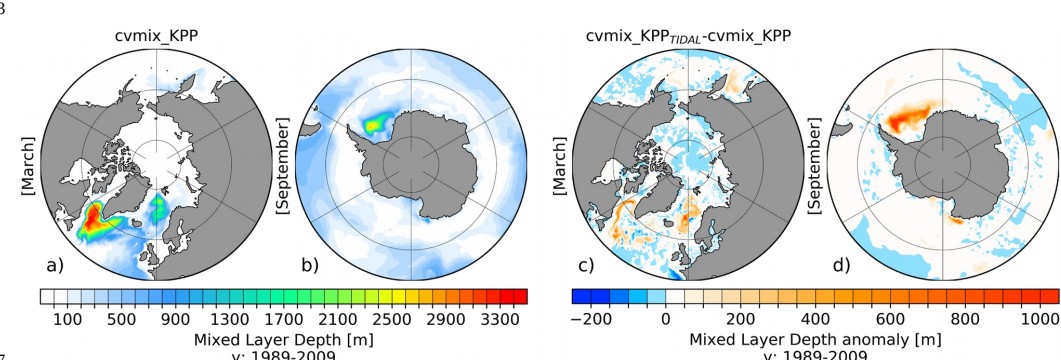

977

**Figure 15:** Northern hemispheric March (a) and southern Hemispheric September (b) mixed layer depth (MLD) for cvmix_KPP without TIDAL mixing as well as corresponding anomalous MLD between cvmix_KPP with minus without TIDAL mixing of Simmons et al. (2004)(c, d), averaged for the period 1989-2009.

982

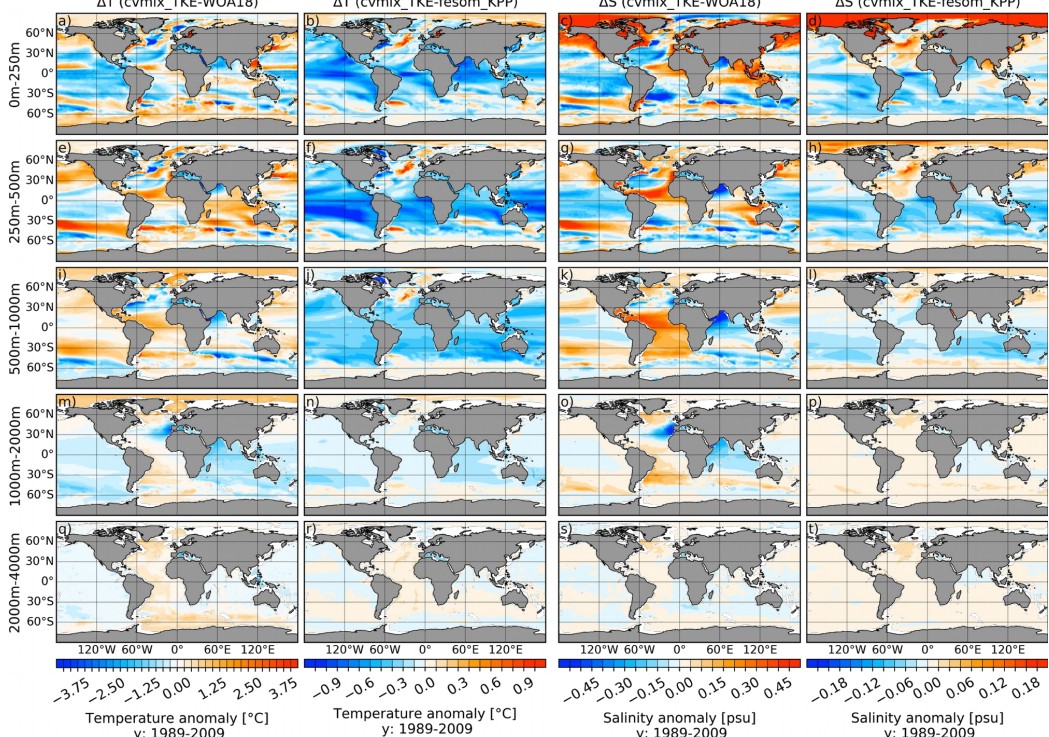

983

**Figure 16:** Temperature (1st and 2nd column), salinity (3rd and 4th column) difference between cvmix_TKE and WOA18 (1st and 3rd column) as well as between cvmix_TKE and fesom_KPP (2nd and 4th column) averaged for the period 1989 to 2009. From top to bottom, panels show the vertically averaged fields for the depth ranges of 0-250 m, 250-500 m, 500-1000 m, 1000-2000 m and 2000-4000 m.

988



75

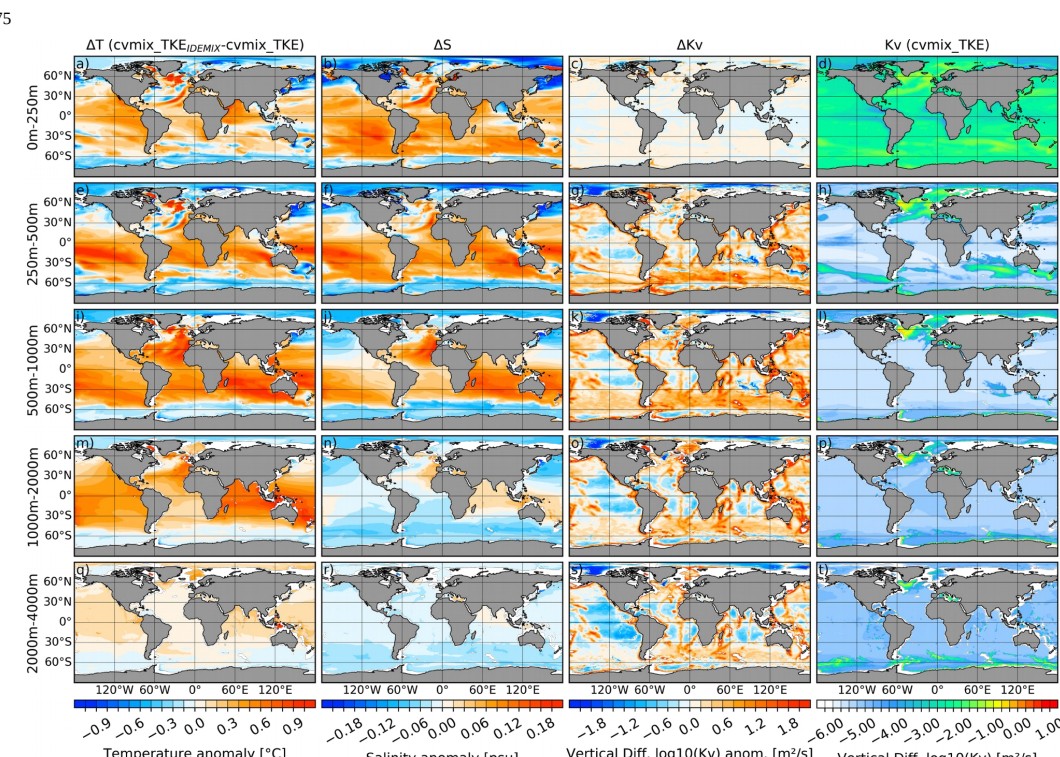

**Figure 17:** Temperature (1st Column), salinity (2nd column) and vertical diffusivity (3rd column) difference between cvmix_TKE with and without IDEMIX as well as the absolute vertical diffusivity values (4th column) for cvmix_TKE without IDEMIX mixing averaged for the period 1989 to 2009. From top to bottom, panels show the vertically averaged fields for the depth ranges of 0-250 m, 250-500 m, 500-1000 m, 1000-2000 m and 2000-4000 m.

77



**Figure 18:** Global zonal averaged biases of temperature (a, b), salinity (c, d) and vertical diffusivity (e) profiles of cvmix_TKE with respect to WOA18 (a, c) and of cvmix_TKE with IDEMIX versus without (c, d, e).





79

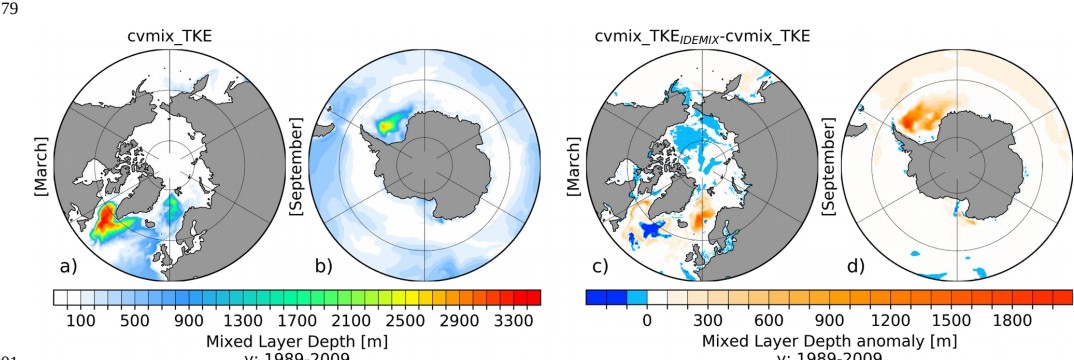

**Figure 19:** Northern hemispheric March (a) and southern Hemispheric September (b) mixed layer depth (MLD) for cvmix_TKE without IDEMIX mixing as well as corresponding anomalous MLD between cvmix_TKE with minus without IDEMIX mixing, averaged for the period 1989-2009.

81

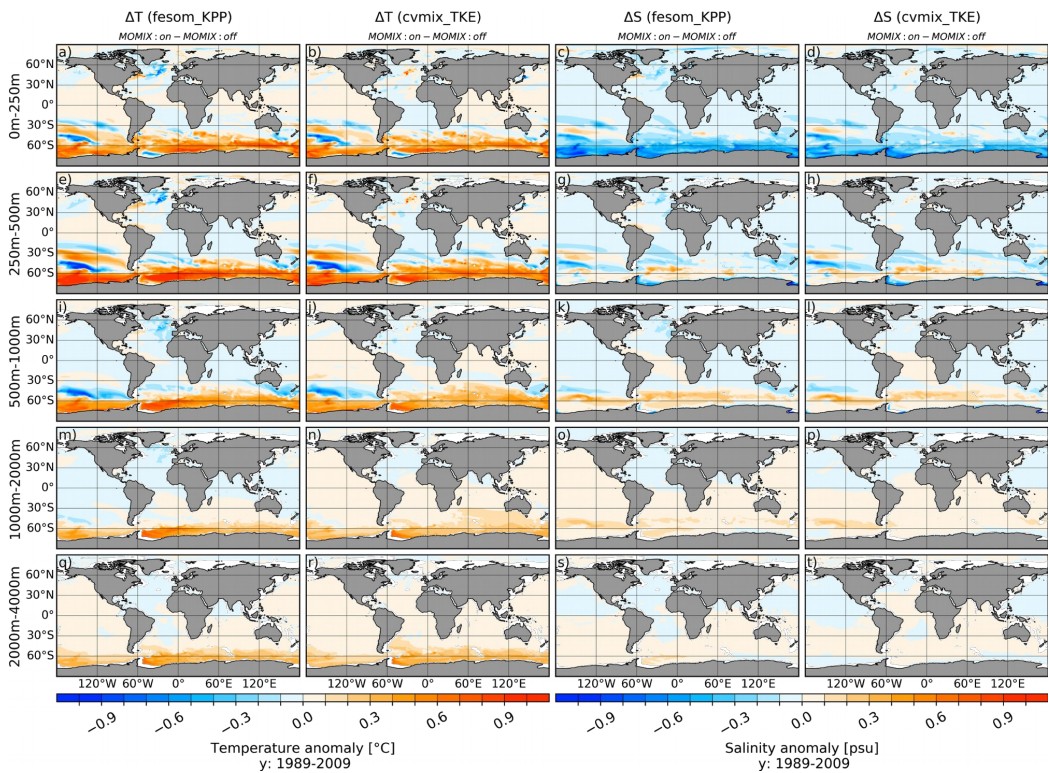

**Figure 20:** Temperature (1st and 2nd column), salinity (3rd and 4th column) difference between fesom_KPP and cvmix_TKE vertical mixing parameterisation with Monin-Obukov vertical mixing (MOMIX) switch on and off (MOMIX: on minus MOMIX: off) averaged for the period 1989 to 2009. From top to bottom, panels show the vertically averaged fields for the depth ranges of 0-250 m, 250-500 m, 500-1000 m, 1000-2000 m and 2000-4000 m.

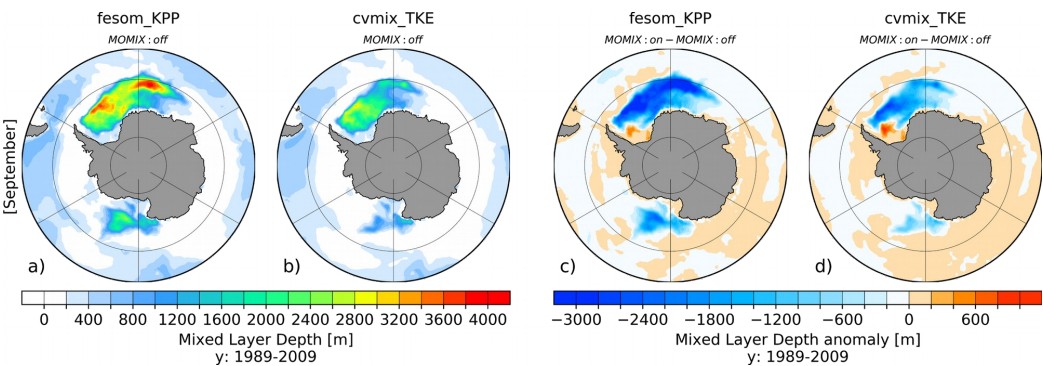

**Figure 21:** Southern Hemispheric September mixed layer depth (MLD) for fesom_KPP (a) and cvmix_TKE (b) with switch off Monin-Obukov vertical mixing (MOMIX) parameterisation as well as corresponding anomalous MLD between switched on and off MOMIX parameterisation (c, d, MOMIX: on minus MOMIX: off), averaged for the period 1989-2009.

83

1020

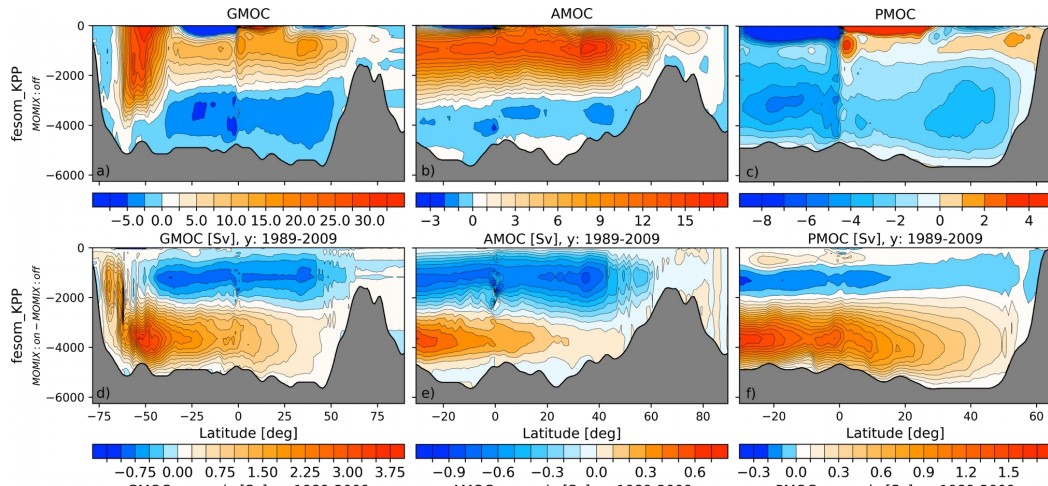

1021

**Figure 22:** Absolute (upper row) and anomalous (lower row) Global (GMOC, left column), Atlantic (AMOC, middle column) and Indo-Pacific (PMOC, right column) Meridional Overturning Circulation, averaged for the time period 1989-2009. Absolute values are shown for fesom_KPP with switched off Monin-Obukov vertical mixing (MOMIX) parameterisation, anomalous values show the difference between fesom_KPP with switch on/off MOMIX parameterisation MOMIX: on minus MOMIX: off).

1027

84

