# Peer review of "Assessment of the Finite VolumE Sea Ice Ocean Model (FESOM2.0), Part II: Partial bottom cells, embedded sea ice and vertical mixing library CVMix"

_Geoscientific Model Development, 2021_

## Referee Comment (RC2)

**REVIEW of**
**" Assessment of the Finite VolumE Sea Ice Ocean Model
(FESOM2.0), Part II: Partial bottom cells, embedded sea ice and
vertical mixing library CVMIX" by**          Scholz et al.,  2021.

The presented manuscript describes and evaluates the application of the recently developed unstructured-mesh Finite-volumE Sea ice-Ocean Model version 2.0 (FESOM2.0).  The current state of the art FESOM2.0 model include multiple options and the authors analyze the impact of the  partial cell, embedded sea ice-ocean model coupling and several mixing parameterizations from the Community Vertical Mixing (CVMIX) project library on the  performance of the coarse resolution global sea ice FESOM2.0 model.

 During the last decades the FESOM model was actively developed and applied for sea ice modeling in multiple publications. Therefore, the detailed analysis of different FESOM2.0 options is a necessary step and will be extremely useful for potential FESOM users. The manuscript is relatively well written and provides many Figures which illustrate the impact of different options on the model solution. Meanwhile, I find that the authors provide only "technical and qualitative comparison" and that makes this manuscript to be very similar to a technical report but not a research paper.

Thus, I definitely may recommend the manuscript for publication in the Geoscientific Model Development Journal, but recommend adding some *quantitative* comparison between different FESOM2.0 options and provide at least a *qualitative physical explanation* of the revealed differences. This is especially important towards  understanding the impact of the partial cells and embedded sea ice options.
Below, I provide my comments and remarks which I would suggest should be taken into account.

**Line 54-55**  *- "Embedded Sea ice … "*

As I remember, the first understanding of the importance of the  Embedded Sea ice was provided by Hibler et al., 1998,  and after that, was used in several publications (e.g. Hutchings et al, 2005). I guess Hibler's embedded ice models are different from FESOM2.0, but these publications are also related to the embedded  sea ice and should be at least cited.

**Line *57*** *"implementation of embedded sea ice relies on the zstar vertical-coordinate option in FESOM2 and also on  the fact that the sea ice component is called on each time step of the ocean model"*

As I understand FESOM2 uses an EVP solver (or its modification) and this suggests the application of two time steps. Which time steps do you mean for the ocean and sea ice models? Also, the modern sea ice model (e.g. CICE6, MIT ocean model)  typically includes explicit and implicit solvers: EVP, different VP solvers (GMRES, Newtonian..)? Do you plan to include an implicit VP solver? If so, it could  be expensive to use a VP solver for each time step of the ocean model.

**Line 151**   *we limited the thickness of the partial bottom cell to be at least half of the full cell layer thickness to reduce the possibility of violating the  vertical Courant–Friedrichs–Lewy (CFL) criterion.*

It looks like a strong limitation: usually the CFL is defined by dz near the surface, so it looks strange to not allow the near bottom dz to be about 0.25 from the full dz. Did you observe any instability in the FESOM numerical scheme?

**Liner 159-184**
*In order to demonstrate the effect of the partial cells on the simulated ocean state we performed two model simulations using the full cell and partial cell approaches, respectively. We investigate, first, the temperature biases of the full cell approach with respect to the data of the World Ocean Atlas 2018 (WOA18, Locarnini et al., 2018; Zweng et al., 2018, in the left column of Fig. 1) and, second, the temperature differences between partial cell and full cell (partial-full) averaged over five different depth ranges 0-250m, 250-500m, 500-1000m, 1000-2000m and 2000-4000m (in the right column of Fig. 1)*

When you compare the FESOM2 performance in this (or similar) figure, you provide only a qualitative comparison ,which sometimes looks subjective. You need to provide some quantitative criteria (e.g. STD from WOA ) for  each (0-250m, 250-500m, 500-1000m, 1000-2000m and 2000-4000m ) layer and include these numbers into the Figures. This will provide real estimates of the positive impact of the partial cells and/or other FESOM2 options.

**Lines: 168-174 + *Figure 1***

This is a very interesting figure, could you please explain the intensive zonal features in Full Cell cases? Note, sometime these features change the sign between 0-250m and 250-500m.
I see some "correlation" with figure 12, from Adcroft et al., 1997, so, some analysis would be very useful.

*Figure 4*

Could you please provide the reference related to the intensive convection in the Weddell Sea and south from Greenland?

Line 216-217 – "*One can summarize that partial cells lead to a clear improvement of the circulation pattern, especially regarding the branch of the Gulf Stream and NAC even in rather coarse resolved configurations.*"

Where can I see that?  Maybe anadditional plot?

**Line 222**   (*linfs*)
(linfs)-> LinFS?

**Line 234**   *"…  reality sea ice and ocean velocities are rarely identical especially in the presence of high frequency wind"*

Not accurate: even if wind is 0, there is a turning angle in the ocean forcing, so, ice will be flowing in a different direction.

**Line 236***; variability especially near the ice-edge where ice divergence/convergence is large (Campin et al., 2008).*

Are you talking about MIZ? In the miz, the concentration is low and ice is especially thin. Not sure that may provide significant impact into the overall sea ice dynamics.

**Line 247;** Figure 6f.
As I see, the embedded ice model provides less ice in the deep Beaufort Sea and more ice in the shallow ESS sea. Could you please provide any explanations?
The same for the Figure 6e – anomaly in the Greenland Sea in March. Why?

**Lines 261- 263?** *The temperature and salinity differences reveal that a significant warming of up to 0.5°C and a salinification of up to 0.10 psu occurs in almost the entire AO due to embedded sea ice, except in a thin stripe along the eastern continental shelf of the AO that shows negative anomalies in the depth ranges of 0-250 m, 250-500 m and 500-1000 m*

*Question:* I always suggested that embedded sea ice  provides a stronger divergence/convergence due to stronger ocean forcing, and that  the embedded sea ice should provide more "open" water or polynyas and that should cool the near surface layer. Am I wrong? Did you estimate the "embedding" impact on the sea ice divergence/convergence?

**Line 265-…** : *The changes in temperature and salinity can be explained by the changes in ocean currents.*

Do I understand this correctly: you explain the temperature/salinity increase due to stronger inflow of the Atlantic water into the AO? If so, you need to estimate this inflow through the Fram strait and provide the numbers.

**Figure 7; -**
You analyze the impact of the embedded sea ice in the AO and Southern Ocean. Why do you need the area between 60S and 60N? It is hard to see any features in the AO. Could you please re-plot this figure and remove the area 60S-60N and increase the AO /Antarctic region?

**Line 393:** "… all topographic features which is induced by the tidal vertical mixing parameterization"

Which topography features? What does it mean ALL? Besides, you may mention the increase of diffusivity mixing in the Indonesian region where internal waves are well-known features of the local dynamics.

**Figure 14**  ….
Something wrong with capture… Check it.

**Figure 15**

I see a strong increase in Weddell Sea in the Antarctica> ? Any evidence /references that tides are active in this area?

**Lines 437-439**
Discussion of figure 18 should be NOT be before the discussion of figure 17.!
Change the order of the figures.

**Line 477 + Figure 17**
Figure 17 should be Figure 18. See above.

**For all figures:**

1)How is it possible that the coastline for 2000-4000m is the same as for 0.50m?  Strange.

2) As I mentioned above,  the disruption of these figures require some numbers which will allow you to estimate the difference between different mixing and numerical schemes utilized in FESOM2.0.  It is reasonable to provide these quantitative differences for each layer.
3)  Physical explanations of the observed differences between different schemes/options is necessary, as well, in most of the cases.

**References:**

Hibler W., P. Heil, and V. I. Lytle, 1998: On simulating high-frequency variability in Antarctic sea-ice dynamics models. Ann. Glaciol., 27, 443–448

Hutchings et al, 2005, Modeling Linear Kinematic Features in Sea Ice, Mon. Wea Rev, 3481-3497, 2005.

---

## Author Response (AR1)

**REVIEW #1**

**"Assessment of the Finite VolumE Sea Ice Ocean Model (FESOM2.0), Part II: Partial bottom cells, embedded sea ice and vertical mixing library CVMIX" by Scholz et al., 2021.**

The presented manuscript describes and evaluates the application of the recently developed unstructured-mesh Finite-volumE Sea ice-Ocean Model version 2.0 (FESOM2.0). The current state of the art FESOM2.0 model include multiple options and the authors analyze the impact of the partial cell, embedded sea ice-ocean model coupling and several mixing parameterizations from the Community Vertical Mixing (CVMIX) project library on the performance of the coarse resolution global sea ice FESOM2.0 model.

During the last decades the FESOM model was actively developed and applied for sea ice modeling in multiple publications. Therefore, the detailed analysis of different FESOM2.0 options is a necessary step and will be extremely useful for potential FESOM users. The manuscript is relatively well written and provides many Figures which illustrate the impact of different options on the model solution. Meanwhile, I find that the authors provide only "technical and qualitative comparison" and that makes this manuscript to be very similar to a technical report but not a research paper.

Thus, I definitely may recommend the manuscript for publication in the Geoscientific Model Development Journal, but recommend adding some *quantitative* comparison between different FESOM2.0 options and provide at least a *qualitative physical explanation* of the revealed differences. This is especially important towards understanding the impact of the partial cells and embedded sea ice options.

Below, I provide my comments and remarks which I would suggest should be taken into account.

*We thank the reviewer for his efforts and constructive comments. We tried to thoroughly include all of his comments or answer his concerns.*

**Line 54-55:** - *"Embedded Sea ice … "*

As I remember, the first understanding of the importance of the Embedded Sea ice was provided by Hibler et al., 1998, and after that, was used in several publications (e.g. Hutchings et al, 2005). I guess Hibler's embedded ice models are different from FESOM2.0, but these publications are also related to the embedded sea ice and should be at least cited.

*We thank the reviewer for the hint to the publications of Hibler et al. 1998 and Hutchings et al. 2005 and will cite them in the manuscript.*

**Line 57:** *"implementation of embedded sea ice relies on the zstar vertical-coordinate option in FESOM2 and also on the fact that the sea ice component is called on each time step of the ocean model"*

As I understand FESOM2 uses an EVP solver (or its modification) and this suggests the

application of two time steps. Which time steps do you mean for the ocean and sea ice models? Also, the modern sea ice model (e.g. CICE6, MIT ocean model) typically includes explicit and implicit solvers: EVP, different VP solvers (GMRES, Newtonian..)? Do you plan to include an implicit VP solver? If so, it could be expensive to use a VP solver for each time step of the ocean model.

***We refer here to the time step of the ocean model, not the sub cycled time steps of the sea ice model. The shown model results use the standard EVP method of Hunke and Dukowicz, 1997 using $N_{EVP}$=150 subcycles. We will consider using a VP solver, but only if we manage to make it as efficient as the EVP solver.***

**Line 151:** *"we limited the thickness of the partial bottom cell to be at least half of the full cell layer thickness to reduce the possibility of violating the vertical Courant–Friedrichs–Lewy (CFL) criterion."*

It looks like a strong limitation: usually the CFL is defined by dz near the surface, so it looks strange to not allow the near bottom dz to be about 0.25 from the full dz. Did you observe any instability in the FESOM numerical scheme?

**The limit of 0.5 was chosen for two reasons: 1st., we wanted to prevent the bottom layer in shallow shelf areas from becoming too small especially in the vicinity of strong boundary currents. It has been found that these regions lead to instabilities of CFL type, which reduce the throughput of the model. Second, in the abyssal ocean we were willing to limit pressure gradient errors. The threshold of 0.5 helps to achieve this. Nevertheless, this limit is a parameter in the model that users can specify depending on their application.**

**Lines 159-184:** *"In order to demonstrate the effect of the partial cells on the simulated ocean state we performed two model simulations using the full cell and partial cell approaches, respectively. We investigate, first, the temperature biases of the full cell approach with respect to the data of the World Ocean Atlas 2018 (WOA18, Locarnini et al., 2018; Zweng et al., 2018, in the left column of Fig. 1) and, second, the temperature differences between partial cell and full cell (partial-full) averaged over five different depth ranges 0-250m, 250-500m, 500-1000m, 1000-2000m and 2000-4000m (in the right column of Fig. 1)"*

When you compare the FESOM2 performance in this (or similar) figure, you provide only a qualitative comparison ,which sometimes looks subjective. You need to provide some quantitative criteria (e.g. STD from WOA ) for each (0-250m, 250-500m, 500-1000m, 1000-2000m and 2000-4000m ) layer and include these numbers into the Figures. This will provide real estimates of the positive impact of the partial cells and/or other FESOM2 options.

***The problem here is that improvements due to partial cells or also other options are mostly local (e.g. limited to the regions of zonal fronts), whereas other regions also can show an increase in bias. Therefore improvements in STD or RMSE on a global scale due to partial cells are rather marginal or even not visible. However we could provide STD or RMSE estimates on a more***

*regional scale e. g. for the North Atlantic (-80<lon<5, 35<lat<70) Gulf Stream and North Atlantic Current region.*

| North Atlantic (-80<lon<5, 35<lat<70) | STD (respect to WOA18) | | RMSE (respect to WOA18) | |
|---|---|---|---|---|
| | PC:0 | PC:1 | PC:0 | PC:1 |
| 0-250m | 1.42 | 1.35 | 1.27 | 1.19 |
| 250-500m | 1.31 | 1.28 | 1.18 | 1.12 |
| 500-1000m | 0.84 | 0.82 | 0.75 | 0.71 |
| 1000-2000m | 0.59 | 0.61 | 0.53 | 0.56 |
| 2000-4000m | 0.48 | 0.50 | 0.48 | 0.49 |

*There, it can be seen that for the upper and intermediate depth ranges in the North Atlantic partial cell leads to an improvement of the STD and RMSE, while the very deep depth ranges indicate a rather marginal increase in STD and RMSE when using partial cells. In the revised manuscript we provide this table in supplementary material.*

**Lines: 168-174 +** *Figure 1*

This is a very interesting figure, could you please explain the intensive zonal features in Full Cell cases? Note, sometime these features change the sign between 0-250m and 250-500m.

I see some "correlation" with figure 12, from Adcroft et al., 1997, so, some analysis would be very useful.

*We are not able to answer this question with the model configuration used here. A simplified test case as the one in Adcroft et al. 1997 would be required for that. Also, the output frequency does not allow us to analyze the excitation of possible gravity waves on the topography steps.*

*Figure 4:* Could you please provide the reference related to the intensive convection in the Weddell Sea and south from Greenland?

*We will refer in the paper to the classical review paper of Marshall and Schott (1999), "open-ocean convection: observation, theory, and models". Intensified deep convection south of Greenland is a known feature in modeled (Danabasoglu et al. 2014) but also observed mixed layer depth (de Boyer Montégut et al. 2004). Mixed layer depth in FESOM is in the range of other ocean models (see Danabasoglu et al. 2014). Intensive convection in the Weddell Sea is also a common feature especially in coarser ocean models (e.g. Sallée, J. B., et al. 2013) although here it is often overestimated due to an*

**underestimation of summertime surface mixing in the southern ocean (Timmermann, R. and Beckmann, 2004).**

**Line 216-217:** "*One can summarize that partial cells lead to a clear improvement of the circulation pattern, especially regarding the branch of the Gulf Stream and NAC even in rather coarse resolved configurations.*"

Where can I see that? Maybe an additional plot?

*We originally wanted here to refer especially to the reduced zonallity of Gulf Stream and NAC when using partial cells. This can be seen in Fig. 1 in the negative temperature biases along the American east coast but also in Fig 3. in the norm of horizontal velocity profiles which indicates, especially for the upper two depth ranges, an enhanced meridionality at 30°W when using partial cells. We changed the text in the revised paper to avoid confusion.*

**Line 222:** (*linfs*)

(linfs)-> LinFS?

*We would like here to stick with the notation of the previous FESOM2 paper of Scholz, P. et al. 2019.*

**Line 234:** "*… reality sea ice and ocean velocities are rarely identical especially in the presence of high frequency wind*"

Not accurate: even if wind is 0, there is a turning angle in the ocean forcing, so, ice will be flowing in a different direction.

*we change the text to "...reality sea ice and ocean velocities are not identical." to avoid confusion.*

**Line 236:** *variability especially near the ice-edge where ice divergence/convergence is large (Campin et al., 2008).*

Are you talking about Marginal Ice Zone (MIZ)? In the miz, the concentration is low and ice is especially thin. Not sure that may provide significant impact into the overall sea ice dynamics.

*We refer here to the seasonal dynamically changing ice-edge which also includes the MIZ. Fig.8 showing the difference in the Arctic ocean circulation with and without embedded sea ice reveals quite some impact on the overall ocean dynamics.*

**Line 247:** Figure 6f.

As I see, the embedded ice model provides less ice in the deep Beaufort Sea and more ice in the shallow ESS sea. Could you please provide any explanations?

The same for the Figure 6e – anomaly in the Greenland Sea in March. Why?

a) Greenland Sea March:

[Figure]

| anomalous sst | anomalous norm of ice velocity | anomalous norm of surface ocean velocity |

It is difficult to analyse this from our data, since we did not save seasonal information for the 3d ocean variables of temperature, salinity, diffusivity and velocity. But from the data we have it looks like that more Sea ice in March in Greenland Sea when using embedded sea ice is related to colder surface ocean, slightly stronger sea ice export from fram strait along sea ice edge and stronger east greenland current out of the Fram Strait.

b) East Siberian Sea in September

For the dipole-like structure in September, with less ice in the deep Beaufort Sea and more ice in the shallow East Siberian Sea the message is less conclusive also due to the fact that we can not provide seasonal information for the 3d variables. The anomalous SST only tells us that there is a colder ocean in the East Siberian Sea and Laptev Sea that allows for more sea ice production and transport within the transpolar drift.

[Figure]

| anomalous sst | anomalous sea surface salinity | |

**Lines 261- 263:** *The temperature and salinity differences reveal that a significant warming of up to 0.5°C and a salinification of up to 0.10 psu occurs in almost the entire AO due to embedded sea ice, except in a thin stripe along the eastern continental shelf of the AO that shows negative anomalies in the depth ranges of 0-250 m, 250-500 m and 500-1000 m*

*Question:* I always suggested that embedded sea ice provides a stronger divergence/convergence due to stronger ocean forcing, and that the embedded sea ice should provide more "open" water or polynyas and that should cool the near surface layer. Am I wrong?

Did you estimate the "embedding" impact on the sea ice divergence/convergence?

***In this part we did not focus on the effect of embedded sea-ice onto the sea ice itself, we focused on the effect for the ocean.***

**Line 265-… :** *The changes in temperature and salinity can be explained by the changes in ocean currents.*

Do I understand this correctly: you explain the temperature/salinity increase due to stronger inflow of the Atlantic water into the AO? If so, you need to estimate this inflow through the Fram strait and provide the numbers.

***The inflow current into the AO gets up to 4 times stronger in a depth range of around 500m and up to 2 times stronger in a depth range of 2000m when using embedded sea ice. We will include these numbers in the revised manuscript.***

[Figure]

| Transport in Sv through section for levitating sea ice (positive eastward transport) | embedded sea ice |
|---|---|
| anomalous volume transport through section (embedded-livitating) | section position |

**Figure 7:**

You analyze the impact of the embedded sea ice in the AO and Southern Ocean. Why do you need the area between 60S and 60N? It is hard to see any features in the AO. Could you please re-plot this figure and remove the area 60S-60N and increase the AO /Antarctic region?

*We agree with the reviewer and will replace the figures with a north polar projection from 30°N-90°N. We do not show the Southern Ocean since the effect of embedded sea ice onto the SO is marginal, which will be stated in the text.*

**Line 393:** "… all topographic features which is induced by the tidal vertical mixing parameterization"

Which topography features? What does it mean ALL? Besides, you may mention the increase of diffusivity mixing in the Indonesian region where internal waves are well-known features of the local dynamics.

*We referred here to the difference in vertical Diffusivity of cvmix_KPP with tidal mixing minus without. We wanted to highlight here that due to the tidal mixing parameterisation of Simmons et al 2004, the vertical diffusivity is increased along the sloping bottom topography like the Midatlantic Ridge or the enhanced mixing in the Indonesian region but also along the continental shelf regions. We will try to better clarify this statement in the text.*

**Figure 14:**

Something wrong with capture… Check it.

*We will try to reformulate this caption.*

**Figure 15:**

I see a strong increase in Weddell Sea in the Antarctica> ? Any evidence /references that tides are active in this area?

*We show here the tidal energy dissipation flux due to bottom drag and energy conversion into internal waves from Jayne and St. Laurent 2001 in the Weddell Sea which also serves as an input parameter for the tidal mixing of Simmons et al. 2004. This clearly shows that there is tidal activity along the continental slope of the Weddell Sea.  Also see A. Foldvik et al.  (1990), "The tides of the southern Weddell Sea", https://doi.org/10.1016/0198-0149(90)90047-Y*

[Figure]

tidal wave dissipation energy

tidal energy (W/m^2)

Data Min = -0,00, Max = 1,41

**Lines 437-439**:

Discussion of figure 18 should be NOT be before the discussion of figure 17.!

Change the order of the figures.

***There is a typo here, it must be: … A closer inspection of temperature and salinity differences between cvmix_TKE and fesom_KPP (Fig. 16, 2nd and 4th column) reveals that cvmix_TKE produces an up to 0.5°C colder ocean...***

**Line 477 + Figure 17:**

Figure 17 should be Figure 18. See above.

***See our reply to the last comment..***

**For all figures:**

1)How is it possible that the coastline for 2000-4000m is the same as for 0.50m?  Strange.

***We always plotted the surface coastline for orientation, the bottom topography for 2000-4000m is left as white (as seen in the absolute  Kv plotts), unfortunately white is also part of some of the colormaps which makes the bottom not very well visible. We will try to highlight the bottom topography with a lighter gray to better highlight it.***

2) As I mentioned above,  the disruption of these figures require some numbers which will allow you to estimate the difference between different mixing and numerical schemes utilized in FESOM2.0.  It is reasonable to provide these quantitative differences for each layer.

*As mentioned above it is difficult to pinpoint the improvement at a single global number, since the improvements are mostly regional where in other regions biases can even grow. Nevertheless we will try to include some numbers for RMSE either on a global or regional scale.*

3) Physical explanations of the observed differences between different schemes/options is necessary, as well, in most of the cases.

*A thoroughly physical explanation of all the observed differences in all the schemes and options might exceed the evaluative and descriptive character of this publication. For some of the observed differences, dedicated studies with more process-orientated sensitivity simulations would be necessary.*

**References:**

Hibler W., P. Heil, and V. I. Lytle, 1998: On simulating high-frequency variability in Antarctic sea-ice dynamics models. Ann. Glaciol., 27, 443–448

Hutchings et al, 2005, Modeling Linear Kinematic Features in Sea Ice, Mon. Wea Rev, 3481-3497, 2005.

**REVIEW #2**

**" Assessment of the Finite VolumE Sea Ice Ocean Model (FESOM2.0), Part II: Partial bottom cells, embedded sea ice and vertical mixing library CVMIX" by Scholz et al., 2021.**

The manuscript of Scholz et al documents a large number of sensitivity tests related to new features for FESOM2 (partial bottom cells, sea ice coupling, and vertical mixing). I was extremely impressed by the vast number of simulated years done for this manuscript and the quality of FESOM developments and the writing. While I found the paper sound and clear for the most part, I have a couple major concerns and more detailed minor comments that I would ask the authors to consider.

***We thank the reviewer for his efforts and constructive comments. We tried to thoroughly consider all of his comments or answer his concerns.***

First and foremost, I struggled in the manuscript to see the scientific impact to the broader community. This manuscript reads as a FESOM technical report and is no doubt very beneficial to that community.

***The scope of the paper is to first provide technical insights to the broader FESOM community about the ongoing FESOM developments but also of course to the broader and general modelling community.***

However, it was very difficult for me to draw out points of interest to the broader modeling community. A few examples, I felt the discussion and results around partial bottom cells (PBCs) and sea ice coupling did not come across as of interest to the broader community, but it is quite possible that having all these results in one place will be a useful reference. You could consider trying to address causes of biases more directly and what the role of the change was in the circulation change, or perhaps consider making a recommendation of best practice configuration for FESOM at the very least. These could be ways to improve the broader impact of this manuscript.

***We agree here with the reviewer that it would be of benefit for the manuscript to give a recommendation for a best practice configuration based on the presented model options in the discussion and conclusion section.***

As of now, for the most part biases are simply noted and then moved on. There are a few exceptions, it is mentioned the way FESOM computes the bulk Richardson number causes the changes in some biases, but plots of boundary layer depth or surface layer average velocity and buoyancy for each method were not plotted.

***Going into a deep analysis of some of the biases, might exceed the scope and length of this publication, which already has quite some length. This is also one of the reasons why things like buoyancy, boundary layer depth etc. haven't been shown. However we try to present results for the KPP ocean boundary layer depth for fesom_KPP and cvmix_KPP within the supplementary material. Also, the cause of some of the biases regarding partial cells could not fully be clarified in this setup and ask for an own examination in a more simplified or idealized configuration.***

Second, I think there were issues with the discussion of vertical mixing with regards to

CVMix. It is stated that FESOM_kpp is configured in a different manner from CVMix, with the surface layer averaging noted as a key difference. However, this is not correct. CVMix leaves a number of choices up to the calling model, amongst them is the velocity and buoyancy difference for the bulk Richardson number. Griffies et al 2015 recommends using 10% of the boundary layer depth as the surface layer average, but this is not within CVMix itself. We further discuss the dependence on model choices in Van Roekel et al 2018.

***We agree with the concerns of the reviewer. We synchronized our implementation with our project partner models MPIOM and ICON-o and they used MOM6 as a template to implement CVMIX KPP. We will clarify this issue in the manuscript.***

As an example, POP chooses the largest shear between a depth and the surface cell making this one step further than what FESOM chooses. It would be interesting if FESOM made other choices in configuration relative to default CVMix, e.g. Monin obukhov/Ekman limiters, matching at the boundary layer base, shape function parameters, etc… and what the impact of these choices might be.

***In the implementation phase of CVMIX KPP we played with the options for the Monin-Obukhov and Ekman limiters since in fesom_KPP they were activated as a default. But both Monin-Obukhov and Ekman limiters had only a minor impact on the solution.***

Relatedly, I think it is important to note that the FESOM KPP choice of the first layer being the surface layer is not physically consistent with KPP (even Large et al 94). Throughout KPP there are built in assumptions regarding the depth of the surface layer (default to 10% of the boundary layer depth) and assuming the first layer as the surface layer is inconsistent.

***The original fesom_KPP implementation was inspired by the KPP implementation in MOM4.***

While it is a valid assumption it is important to point out this issue and the consequences of it. It would be interesting to explore the impact of this choice, but that is likely beyond the scope of this paper. Here it shows basic plots of T/S, but would be good to know a finer scale view too. Have you conducted a simulation that uses the CVMix library but the FESOM_KPP choice for the numerator and denominator of the bulk Richardson number? This could clearly show differences associated with the choice. My expectation is that the shear is the more dominant term and using the surface value will deepen boundary layers, but it would be interesting to see clear evidence of this. Again, this is a possible place where you could make broader impacts.

To help clarify and grasp the broad points of the paper, I would suggest perhaps having bullet point take summary somewhere (perhaps the discussion/conclusions) with call outs to key figures. A few other suggestions to help with clarity: (1) a table of differences in KPP vs PP and in CVMix / FESOM versions would help maintain clarity (2) a table of the FESOM KPP configuration, e.g. is matching utilized?

**Minor comments:**

Throughout, you write CVMIX, the acronym is CVMix.

In numerous places you have things like "southern hemisphere September" which reads odd to me. I'd suggest parentheses around the month.

***We will change this in the manuscript.***

There are also a number of references that need proofing, e.g. Griffis 2015 and Ilicack2006.

***We will change this in the manuscript.***

There are also many places with subscripts that didn't typeset correctly.

***Originally we tried to avoid subscripts, since they become rather small in the figures. We also tried to avoid the use of hyphens for the description of our experiments since they could be miss interpreted as minus signs, therefore we decided to use an underscore for the description of our experiments (e.g fesom_KPP, cvmix_TKE ...). If this typeset is not wanted, it could be changed.***

**Line specific Comments:**

**Line 27:** – Delete "The" → ***will be corrected.***

**Line 29:** – suggest adding "southern hemisphere" to "sea ice melt season mixing" → ***will be corrected.***

**Line 40 and L41:** remove "one" after "first" and "second" → ***will be corrected.***

**Line 50-57:** – by the word "embedded" I expected the sea ice code to be in the ocean code as in MOM6/SIS, is this the case for FESOM?

***The fact that the sea ice code is within the ocean code is anyway the case for FESOM2, but the term embedded here refers to the publication of Campin et al. 2008 whos came up with the naming and describes the case where the sea ice is embedded in the surface ocean and swims according to its density in the ocean by replacing water instead of only levitating on top of the ocean***

It is not clear. It may be better to say "non-levitating" or "pressure exerting" for clarity, but no need to change the word if clearly defined if the ice code is in the ocean model or uses a coupler.

***We do not use any coupler for the sea ice***

**Line 57:** – you mention that you must compute sea ice at every ocean time step to "embed", this doesn't seem desirable and is not actually required in our experience in MPAS/E3SM. Embedding is more dependent on the fidelity of the ice-ocean coupling in our experience. Can you clarify what you mean by this statement?

***It's no where written that we "must" compute sea ice at every timestep its only written that we rely on zstar for embedded sea ice, which is the case. Otherwise FESOM2 calls in the moment the sea ice routine at each ocean timestep. If this is really necessary or it's sufficient to be called at every second or tenth time step will be evaluated in the future to maybe further close***

*critical bottlenecks.*

Line 81: – It isn't clear to me what "prime vertical mixing" means, is this default?

*For us "prime vertical mixing" schemes refers to schemes that set the general global ocean wide vertical diffusivity like PP, KPP or TKE. "non prime vertical mixing" focus for us on certain processes that are added to the prime vertical mixing, like the breaking of tidal induced internal waves (Simmons et al. 2004, IDEMIX) or local mixing processes like in the Monin-Obukov mixing (MOMIX).*

Line 82: – the phrase "deliver a usable mixing scheme" is confusing to me. Do you have a meaning in mind?

*"Deliver" stands here in the meaning of "to create", " to build" a mixing parameterisation that has certain validity for the entire global ocean. Exchange: ...others that have the purpose to create a general  mixing parameterisation for the ...*

Line 129: – delete the comma and which → *will be corrected.*

Line 145: – It is unclear to me why Pacanowski and Gnanadesikan 98 is discussed. It seems FESOM uses Schepetekin 2003 instead.

*Pacanowski and Gnanadesikan, 1998 discusses first the basic concept of partial cells, how to compute and how to use them. That's why it's worth mentioning them. Schepetekin 2003 provides an alternative way on how to compute the pressure gradient force that was more beneficial for us.*

Line 151-153: – Have you tested FESOM without the requirement that the bottom thickness be greater than ½ the layer thickness? As an example, MPAS-O runs stably without this requirement, but I have not looked in depth at the possible biases that may exist even though it is stable. It could be interesting to further examine this choice.

*We tested mostly for the condition that the layer thickness can not be smaller than half the layer thickness which turned out to be important especially in the shallow shelf areas to keep the time step in limit and to avoid critical vertical CFL conditions. Anyway MOM6 is using this fully lagrangian layer motion with the subsequent remapping, where critical CFL conditions are not an issue anymore. In FESOM2 we will explore this option in the near future.*

Line 157: – vertice -> vertex → *will be corrected.*

Line 165: – biasin -> bias in → *will be corrected.*

Section 3.2 – In this section you discuss a dependence on sea ice thickness in configuration choices but then only present comparison to sea ice concentration. It would be helpful to plot thickness. This is also more consistent with what is actually measured by satellite.

*The results between sea ice concentration and thickness are rather similar and there exist more observational derived sea ice concentration estimates (e.g. NSIDC) than thickness estimates, especially with respect to long term climatology. Nevertheless, we add plots for the absolute sea ice thickness and anomaly with the supplementary.*

**Line 281:** – while MOM6 does have a branch with CVMix the original implementation was designed to reproduce the POP formulation, so I would change MOM6 to POP → ***will be corrected.***

**Line 315+:** - the strong similarity between KPP and PP in the analysis was surprising to me. These schemes are quite different, especially in the near surface. Have you examined boundary layer differences? E.g. Mixed layer depth? I wonder if perhaps the similarities are due to the fields presented, you show averages over fairly thick layers and below the boundary layer I imagine FESOM uses the shear instability induced mixing of Large et al. 1994, it is possible your analysis is only highlighting more deep ocean impacts and the similarities in the LMD SI induced mixing and PP81 mixing make the results seem similar. A simple test would be comparison of MLD between the schemes.

***There is a considerable difference between KPP and PP, although the climatological biases with respect to WOA18 is still much larger see Suppl. 2. These differences between KPP and PP can be also seen in the Mixed Layer Depth. We added also this figure to the supplementary.***

[Figure]

| ***March MLD fesom_KPP minus fesom_PP*** | ***September MLD fesom_KPP minus fesom_PP*** |

**Line 398:** – ando -> and→ ***will be corrected.***

**Line 450-453:** – Why not test different background options? Seems like a very easy test to do.

**Line 486-488:** – this sentence is very confusing to me. When you use 'except' but discuss freshening in one part and temperature in the other it doesn't read easy to me.

***There is a typo: it's not supposed to be "warming". Exchange: "...The depth ranges below indicate a predominant general freshening almost everywhere, except for the Mediterranean outflow and Indian Ocean which indicate a slight*** ***salinification*** ***...."***

**Line 527-529:** – have you tested combinations of changes? It seems possible (perhaps likely) that some changes have nonlinear interactions and are not as simple as just adding biases.

*We added another figure (Suppl. 9) to the supplementary addressing the improvement in MLD for cvmix_TKE$_{IDEMIX}$ with and without parital cell. It seems that the interaction in this case seems to be rather linear. So the improvement of Weddell Sea circulation seems to be a rather solid feature oof partial cells.*

**Line 549-552:** – any ideas why you see a large change in the gulf stream for the MOMIX + KPP?  Is this related to changes in transport (AMOC maybe?)

*Fig. 22 shows that with MOMIX the upper and lower AMOC cells become weaker. The weakening of the upper cell leads to a weaker meridional heat transport through Gulf Stream and NAC and could lead to the displayed cooling in the North Atlantic with MOMIX. We will add this explanation to the manuscript.*

**Line 582-583:** – As an MPAS-O developer I confess I agree with your statement here, I'm always deeply impressed by the pace and quality of FESOM developments.

*We see MPAS as a rather close competitor that catches up very quickly.*

**Line 701-702:** – add commas around "and to a decrease in the high-latitude" → *will be corrected.*

**Fig. 14 and 18:** – the plot titles seem wrong in most panels here.  Also in panel (c ) of both there is an odd high salinity bias 40N.  It is interesting that it is identical in both Fig 14 and 18.  Is this a plotting or analysis artifact?

*These are not the plot titles, these are the description labels of the colorbar, it might be beneficial to insert here a vertical gap to emphasize this. The high salinity bias at 40° is indeed a bug in the computation of the anomaly. These figures will be corrected and replaced in the manuscript.*

---

## Referee Report (RR1)

**Second REVIEW of**
**" Assessment of the Finite Volume Sea Ice Ocean Model**
**(FESOM2.0), Part II: Partial bottom cells, embedded sea ice and**
**vertical mixing library CVMIX" by                Scholz et al.,  2021.**

In the revised version of the manuscript, the authors made significant progress and carefully addressed most of my questions/comments from the previous review. In particular, they provided some quantitative analysis which is summarized in the Table now. The authors also improved the quality of the Figures and Tables. I am satisfied with most of the  replies provided by the Authors and I think that this manuscript can be published in JTECH after addressing a few minor comments which I provide below:

Minor Question:

1. (old question related to the Line 57:

**Former Line *57:* “implementation of embedded sea ice relies on the zstar vertical-coordinate option**
*in FESOM2 and also on the fact that the sea ice component is called on each time step of the ocean model”*

***We refer here to the time step of the ocean model, not the sub cycled time steps of the sea ice model. The shown model results use the standard EVP method of Hunke and Dukowicz, 1997 using NEVP=150 subcycles. We will consider using a VP solver, but only if we manage to make it as efficient as the EVP solver.***

Ok. If so,   I guess:

a) then that should be mentioned somewhere around line 57.
b) Are 150 iterations enough?  As far as I know, the last tendency is to increase  the number of subcycle iterations up to 2000,  since  “Too small $N_{EVP}$ may lead to numerical noise (see, e.g., Bouillon et al., **2013**; Lemieux et al., **2012**; Losch & Danilov, **2012**)”  https://agupubs.onlinelibrary.wiley.com/doi/full/10.1029/2018MS001485.

This should be discussed.

**Line *59:*** zstar-> z-star

**Line 153-155 (Former Line 1*51):***

*“Furthermore, we limited the thickness of the partial bottom cell to be at least half of the full cell layer thickness to reduce the possibility of violating the vertical Courant–Friedrichs–Lewy (CFL) criterion.”*

I guess, your explanations should be included into this sentence, somehow.  For example, mention that this limitation is for shallow regions only …

**Former line 265**

Thank you for providing an explanation and, especially, for the volume transport figure. Actually, I like this figure very much and suggest including it into the Supplemental material! The inflow of the warm AW into Arctic Ocean is the key question in Arctic Ocean modeling and this result may be extremely useful for Arctic Ocean modelers.

---

## Author Response (AR2)

**Comments to the author**:**

To the authors,

Thank you for your thoughtful revisions and patience throughout the review process. Both reviewers agree that the revised version is ready for publication and therefore I am happy to accept the paper for publication pending a few small final revision (as can be found in the most recent round of reviews). Provided you can make these corrections in a timely manner, we can proceed with the publication process.

Best,

Alex Robel

Topical Editor, GMD

**Reviewer 1**:**

The revision addresses all of my concerns and upon a second review I believe my initial assessment about the broad applicability of this work was not fully justified. I agree with the author's response that the highly detailed experiments especially the mix and match vertical mixing runs will be a nice baseline for other modeling centers to compare against. Further, the manuscript is well structured and well written. Thus I recommend this to be accepted.

I only have one very minor technical issue. In a few places, e.g. L458, the phrase 'OBLd is enhanced' is used. I would suggest deepens instead of enhanced.

*We corrected this in the manuscript*

**Reviewer 2:**

Second REVIEW of

" Assessment of the Finite Volume Sea Ice Ocean Model (FESOM2.0), Part II: Partial bottom cells, embedded sea ice and vertical mixing library CVMIX" by Scholz et al., 2021. In the revised version of the manuscript, the authors made significant progress and carefully addressed most of my questions/comments from the previous review. In particular, they provided some quantitative analysis which is summarized in the Table now. The authors also improved the quality of the Figures and Tables. I am satisfied with most of the replies provided by the Authors and I think that this manuscript can be published in JTECH after addressing a few minor comments which I provide below:

Minor Question:

1. (old question related to the Line 57:

Former Line 57: "implementation of embedded sea ice relies on the zstar vertical-coordinate option in FESOM2 and also on the fact that the sea ice component is called on each time step of the ocean model"

We refer here to the time step of the ocean model, not the sub cycled time steps of the sea ice model. The shown model results use the standard EVP method of Hunke and Dukowicz, 1997 using NEVP=150 subcycles. We will consider using a VP solver, but only if we manage to make it as efficient as the EVP solver.

Ok. If so, I guess:

a) then that should be mentioned somewhere around line 57.

*We added this to the manuscript.*

b) Are 150 iterations enough? As far as I know, the last tendency is to increase the number of subcycle iterations up to 2000, since "Too small NEVP may lead to numerical noise (see, e.g., Bouillon et al., 2013; Lemieux et al., 2012; Losch & Danilov, 2012)" https://agupubs.onlinelibrary.wiley.com/doi/full/10.1029/2018MS001485.
This should be discussed.

*There is a publication of Koldunov et al. 2019 which deals with this topic in FESOM2. The outcome was that in coarse resolved configuration 150 subcycles can be sufficient, beyond that the ice model does not significantly improve anymore. However in higher resolved configurations (e.g. ~4.5 km ) significantly more subcycles are necessary to converge to a satisfying solution.*

Line 59: zstar-> z-star

*We would like to stick here with the used notation of zstar, since we used the same in the previous publication.*

Line 153-155 (Former Line 151):
"Furthermore, we limited the thickness of the partial bottom cell to be at least half of the full cell layer thickness to reduce the possibility of violating the vertical Courant–Friedrichs–Lewy (CFL) criterion."

I guess, your explanations should be included into this sentence, somehow. For example, mention that this limitation is for shallow regions only …

*We added this to the manuscript.*

Former line 265
Thank you for providing an explanation and, especially, for the volume transport figure. Actually, I like this figure very much and suggest including it into the Supplemental material! The inflow of the warm AW into Arctic Ocean is the key question in Arctic Ocean modeling and this result may be extremely useful for Arctic Ocean modelers.

*Figures have been added to Supplementary.*